# Tough asymmetric thermochromic ionogels via dynamic in situ phase separation for dual-modal smart optical switching

Guoli Du, Jianing Li, Changxing Wang, Jianing Li, Yayun Ning, Yifan Yue, Yuechi Xie, Sen Yang ✉ & Xuegang Lu ✉

Despite significant advances in thermochromic materials (TCMs) for smart optical switching, achieving simultaneous optimization of optical switching, mechanical robustness, and environmental tolerance remains a critical challenge for their real-world implementation. Herein, we present tough asymmetric thermochromic ionogels (ATIs) fabricated via dynamic in situ phase separation. The ATIs integrate a thermoresponsive light-scattering layer (LCST: 28 °C to 41 °C) and a mechanically reinforced supporting layer through sequential free-radical polymerization. The resulting material exhibits high transparency (>85% at 20 °C), ultralow-temperature resistance (transparent at −70 °C, even at −196 °C), and exceptional toughness (tensile strength >5 MPa, toughness >17 MJ m$^{-3}$). Leveraging hydrophobic ionic liquid as solvent, the ATIs achieve reversible optical switching (transmittance <10% at 40 °C) without encapsulation, with autonomous adhesion (>400 N m$^{-1}$ peel strength on glass). Applied to smart cooling windows, ATIs reduce solar radiation by 56.8% while enabling aesthetic customization. Combined with indirect/direct Joule heating technology, ATIs enable active optical switching and even dynamic projection display, offering a scalable platform for all-weather adaptive optical systems.

The growing demand for adaptive intelligent systems—from advanced sensing, soft robotics to energy-efficient buildings—drives the development of smart materials towards emulating biological "sense-feedback-actuate" mechanisms for self-sensing and active regulation[1]. Among these, stimulus-responsive materials serve as foundational building blocks[2], dynamically altering optical, mechanical, electrical, or morphological properties in response to external triggers including light, pH, electric fields[3,4], humidity, and temperature[5-7]. Thermochromic materials (TCMs), which modulate transparency/color via temperature changes[8], show promise for applications such as visual sensing[9], intelligent actuators[10], and information encryption[11], particularly for smart optical switching—a key technology in next-generation adaptive windows[12], displays, and anti-counterfeiting systems[13].

Existing TCMs—including inorganic compounds ($VO_2$, $WO_3$, halide perovskites)[14-16], organic liquid crystals (cholesteric phase)[17], and nanocomposites (metal nanofillers, quantum dots)[18]—enable reversible light transmission modulation but suffer from high transition temperatures (>60 °C), intrinsic rigidity, and complex fabrication processes, which limit their ability to meet emerging requirements for tunable transition temperature ranges, mechanical toughness, and on-demand responsiveness[19]. Thermochromic polymers (TCPs) offer a promising solution, bridging attributes of organic/inorganic materials[20]. TCPs are typically realized by incorporating thermosensitive polymers (e.g., poly(N-isopropylacrylamide) (PNIPAM), hydroxypropyl cellulose (HPC), poly(propylene oxide) (PPO), and polyurethane (PU) derivatives) into liquid matrices to form gels/blends[21,22]. Like inorganic TCMs, TCPs exhibit rapid phase transitions between transparent and opaque states at lower (LCST) or upper

School of Physics, MOE Key Laboratory for Nonequilibrium Synthesis and Modulation of Condensed Matter, Xi'an Jiaotong University, Xi'an, China.
✉e-mail: yangsen@mail.xjtu.edu.cn; xglu@mail.xjtu.edu.cn

critical solution temperatures (UCST), providing exceptional processability, tunable mechanics, and functional compatibility for advanced thermal-responsive devices. Yet, conventional TCPs like PNIPAM hydrogels—known for their favorable LCST (28–35 °C)[23]—suffer from inherent drawbacks: low crosslinking and high water content yield weak tensile strength (<10 kPa), poor water retention, and sub-0 °C failure[24], necessitating rigid encapsulation that sacrifices flexibility and reprocessability in practical implementation. Although large-scale production is feasible[25], post-polymerization purification involving chain transfer remains a critical barrier to widespread adoption. Efforts to modulate LCST and enhance performance (e.g., random copolymerization, interpenetrating networks, additive doping) have yielded only limited improvements[26,27]. Ionic liquids (ILs) offer a promising avenue for high-stability TCPs due to their compatibility with polymers, thermal stability, and non-volatility. In recent years, Yan and colleagues have systematically optimized the supramolecular structures of IL-based functional materials, leading to the development of a series of high-strength and high-toughness ionogels. Remarkably, the mechanical performance of these ionogels surpasses that of most metals and alloys[28,29], providing a solid foundation for fabricating mechanically robust ILs-based TCPs. Comprehensive reviews by Men et al. summarize the progress in LCST- and UCST-type thermosensitive ionogels, highlighting their role in addressing the limitations of conventional hydrogels and organogels[30,31]. Owing to the low glass transition temperatures and tunable supramolecular characteristics of ILs[32], TCP-based ionogels are expected to maintain stable thermoresponsive behavior under complex and variable temperature conditions. For instance, IL-based thermochromic ionogels fabricated using modified PU/polyvinylidene fluoride (PVDF) matrices exhibit a tunable LCST between 20 and 40 °C, along with robust mechanical properties and improved thermal stability[33]. However, challenges such as complex synthesis routes, stringent processing conditions, and high production costs currently hinder their large-scale commercialization. Therefore, facile fabrication strategies for high-performance TCPs remain a significant challenge, necessitating a deeper understanding of the dynamic phase separation mechanisms governing IL-polymer network interactions—a fundamental prerequisite for optimizing ionogel-based TCPs and advancing next-generation smart thermal-responsive systems.

To achieve a balance between smart responsiveness and mechanical toughness, this study develops an asymmetric ionogel that merges thermochromism with high mechanical strength. Through sequential free-radical polymerization, we implement a synergistic strategy combining thermosensitive dynamic phase separation (reversible) and in situ phase separation (irreversible) to covalently conjugate a thermochromic ionogel layer with a high-strength ionogel layer. The resulting material achieves an optimal LCST of 34 °C, exceptionally low temperature stability (maintaining transparency down to −70 °C and even −196 °C), and robust mechanical properties (tensile strength >5 MPa, elongation >800%, toughness >17 MJ·m⁻³). Hydrophobic monomers/ILs and abundant polar groups endow the ionogel with superior hydrophobicity (water contact angle >135°) and intrinsic self-adhesion (90° peel strength >400 N m⁻¹), enabling direct adhesion to working panels without encapsulation and bonding while maintaining reliable functionality under harsh environmental conditions (hot, cold, dry, and rainy). Beyond enabling passive optical modulation for smart windows, this multifunctional ionogel serves as an active optical switching platform, facilitating local projection displays via indirect heating and acting as large-area projection screens under direct Joule heating.

## Results

### Design strategy of tough asymmetric thermochromic ionogels

This asymmetric thermochromic ionogel (ATIs) with exceptional mechanical robustness integrates: i) a thermoresponsive light-scattering ionogel layer engineered via thermosensitive dynamic phase separation (a temperature-mediated reversible process), and ii) a high-strength ionogel supporting layer fabricated by in situ phase separation (irreversible process)—the two covalently bonded at the interface through sequential photopolymerization (Fig. 1a, Supplementary Fig. 1). The resulting ATIs exhibit not only sensitive thermochromism and high mechanical strength but also remarkable freeze-resistance and tunable hydrophobicity, enabling reliable operation under all-weather conditions. Specifically, the dynamic phase-separated homopolymer (light-scattering ionogel layer, denoted ATI-B) was formed via free radical polymerization of n-butyl acrylate (BA) in the hydrophobic IL 1-ethyl-3-methylimidazolium bis(trifluoromethylsulfonyl)imide ([Emim][TFSI]) (Fig. 1b). Concurrently, the high-strength layer consists of in situ phase-separated P(BA-co-NIPAM) copolymers (denoted ATI-BN) synthesized from BA and NIPAM monomers. Both layers share the same IL solvent, photoinitiator, and crosslinker (Supplementary Fig. 2). The resulting ATIs exhibit high optical transparency and mechanical integrity at room temperature (below LCST) (Supplementary Fig. 3), combining flexibility and impact resistance—properties unattainable in conventional high-strength transparent materials like glass—while demonstrating observable thermochromic performance (Fig. 1c).

Environmental tolerance, especially to high and low temperatures, is an important evaluation index for TCPs[34]. The ATI maintains high transmittance exceeding 80% at room temperature (below LCST) (Supplementary Fig. 4), which is essential for real-world implementations. Upon exposure to temperatures above the LCST—whether through ambient heating or intrinsic Joule heating—it exhibits rapid thermochromic behavior, with transmittance dropping to below 10% at 40 °C (Fig. 1d–i, Supplementary Fig. 5, Movie 1). Owing to the IL solvent matrix, the ATI retains high optical transparency even under low temperature conditions at −70 °C (Fig. 1d–ii), and demonstrates negligible visible-light scattering at liquid nitrogen temperatures (−196 °C) (Supplementary Movie 2), confirming exceptional anti-frost capability. A comprehensive performance comparison between ATIs and reported TCMs was conducted, evaluating transparency, low-temperature resistance, self-supporting (without additional encapsulation), stretchability, solar modulation capability, and self-adhesion (Fig. 1e)[21,35–38]. Notably, ATIs demonstrate superior low-temperature stability and mechanical strength while maintaining their solar modulation capability. For practical applications, the study critically validates ATIs' optical modulation efficacy for smart cooling windows. Beyond this, ATIs are further demonstrated as an active optical switching platform—enabling localized projection displays via indirect heating and global projection screens through direct Joule heating (Fig. 1f, Supplementary Fig. 6). This work not only expands the application scope of ATIs in energy-efficient smart windows but also establishes a preliminary reference paradigm for next-generation display systems based on thermally responsive optical switches.

### Asymmetric structure achieved via dynamic in situ phase separation

Sequential free-radical polymerization was employed to prepare ATIs. Specifically, ATI-B was prepared by one-step photopolymerization, while ATI-BN was in situ photocured based on the former, resulting in the asymmetric Janus structure of ATIs (Supplementary Fig. 7). The hydrophobic monomer BA demonstrates good compatibility with the solvent [Emim][TFSI], forming a homogeneous TCP system through ion-dipole interactions. In contrast, NIPAM is less compatible in imidazolium-based ILs and requires the formation of ATI-BN precursors with assisted solvation of BA (Supplementary Figs. 8 and 9). The ion-dipole interaction sites between the monomer and the solvent in ionogels are given by electrostatic potential (ESP) simulations (Fig. 2a). Fourier transform infrared spectroscopy (FTIR) shows that both homopolymer PBA and copolymer P(BA-co-NIPAM) form ion-

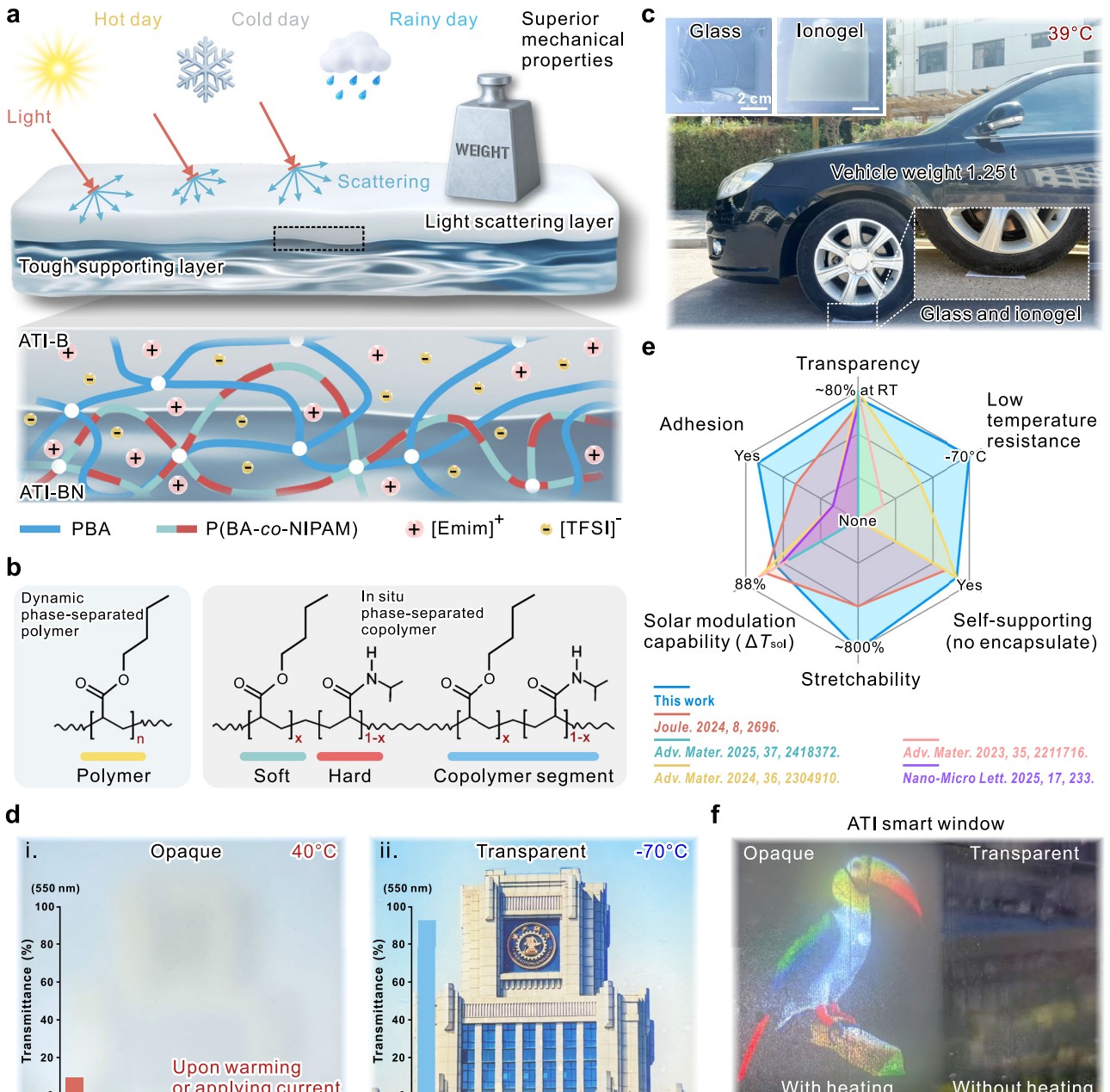

**Fig. 1 | Design strategy and performance of ATIs. a** Schematic of the cross-section of an environmentally adaptable ATI in which the top layer serves as a light scattering layer and the bottom layer serves as a tough supporting layer. **b** Two polymer networks in asymmetric ionogels. Left, dynamic phase-separated polymer network in ATI-B; right, in situ phase-separated copolymer network in ATI-BN. **c** Demonstration of the tough mechanical properties of ATI; inset shows glass (left) and ATI (right) after being crushed by a 1.25 t vehicle. **d** Optical transparency of ATIs in extreme cold (−70 °C) and common high-temperature environments (ambient warming or Joule heating, 40 °C), respectively; the inset shows transmittance of light at 550 nm wavelength. **e** A comprehensive performance comparison between ATIs and reported TCMs in multiple dimensions, including transparency, low-temperature resistance, self-supporting, stretchability, solar modulation capability, and adhesion. **f** ATI as a flexible optical switch mounted on glass: locally heated region (left) serves as a projection screen (low transmittance, a bird image serves as a projection object), while the ambient-temperature region (right) maintains high transparency for unobstructed visibility.

dipole interactions with the solvent [Emim][TFSI] to stabilize the intrinsic gel state (Fig. 2b, Supplementary Fig. 10, Note 1). The ATI-BN at 3380 cm⁻¹ attributed to the -N-H stretching vibration confirms the formation of intramolecular hydrogen bonds during polymerization of NIPAM due to its high incompatibility with the solvent (Supplementary Fig. 11). The cross-section scanning electron microscopy (SEM) image of the homopolymer ATI-B exhibits a smooth homogeneous morphology; whereas ATI-BN shows a rough morphology with peak-like protrusions (Fig. 2c). The reason for this is that the low compatibility of PNIPAM within [Emim][TFSI] causes the formation of a bicontinuous network of alternating polymer-rich and solvent-rich phases within the ionogel (Supplementary Fig. 12)[39]. Meanwhile, the energy dispersive spectroscopy (EDS) results confirm the asymmetric distribution of the elements in ATI (Supplementary Fig. 13).

The in situ and dynamic phase separation in ATIs were analyzed at the microscopic level using small-angle X-ray scattering (SAXS) and wide-angle X-ray scattering (WAXS) techniques. The SAXS results show that ATI-B is highly homogeneous on the 1-50 nm scale, and the absence of scattering rings in the 2D SAXS spectrum proves that there is almost no internal phase separation at room temperature (RT)

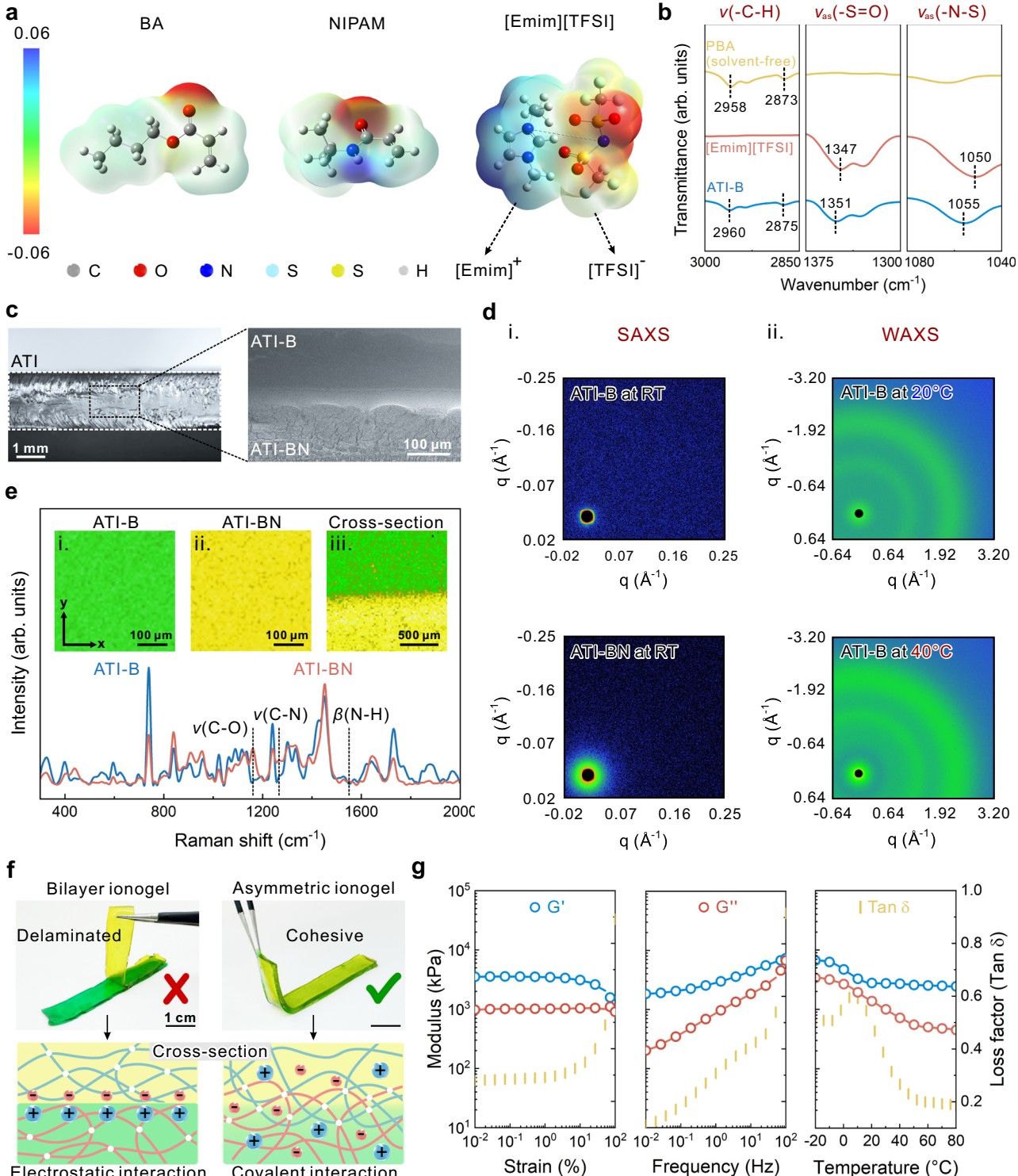

**Fig. 2 | Preparation, composition, and characterization of asymmetric structures of ATIs. a** ESP simulations of key ATI components: polymer monomers BA and NIPAM, and IL solvent [Emim][TFSI]. **b** FTIR spectra of solvent-free PBA, pure IL, and ATI-B. **c** Microscopic image (left) and SEM image (right) of the cross-section of the ATI, distinctly showing its asymmetric structure composed of ATI-BN and ATI-B. **d** 2D spectra of SAXS (i) and WAXS (ii). **e** Raman spectra of ATI-B and ATI-BN; insets i, ii, iii are Raman mapping images of ATI-B, ATI-BN, and ATI cross sections, respectively. **f** Optical images of physical bilayer ionogel and ATI (upper PBA ionogel layer, stained with metanil yellow; lower P(BA-co-NIPAM) ionogel layer, stained with brilliant green) and schematic of the proposed interface. **g** Characterization of the rheological properties of ATIs, including amplitude sweeping (left), frequency sweeping (middle), and temperature ramping (right). G′ is the storage modulus, G″ is the loss modulus, and Tan δ is the loss factor.

(Fig. 2d-i). In contrast, ATI-BN exhibits more pronounced scattering rings, and its SAXS spectrum displays distinct step-like peaks. Through computational fitting, its phase separation size was determined to be approximately 7.3 nm (Supplementary Fig. 14)[40]. This size is far below the visible light wavelength range, so ATI-BN remains highly transparent. The thermally induced transparent-to-opaque transition of ionogels is solely related to ATI-B. The variable-temperature WAXS results of ATI-B reveal that the characteristic dimensions of its scattering peaks shift from 4.5 Å and 7.7 Å to 4.6 Å and 7.2 Å (Fig. 2d-ii, Supplementary Fig. 15 and Table 1). This change can be attributed to the increase in polymer chain spacing due to thermal expansion effects, coupled with the entropy-driven contraction of [Emim]$^+$–[TFSI]$^-$ ion clusters[41]. To further characterize the asymmetric structure of ATI, spatial Raman spectroscopy was employed. The intensity of the absorption bands of ATI-BN at 1160 cm$^{-1}$, 1267 cm$^{-1}$, and 1550 cm$^{-1}$ is significantly stronger than that of ATI-B, which shows the differences in their components (Fig. 2ei-ii). The close and seamless connection between the two phases in the asymmetric structure of ATI can be visualized by 2D Raman mapping of the cross section (Fig. 2e-iii)[42]. Physical bilayer gels prepared by direct electrostatic adhesion are characterized by weak electrostatic interactions between the interfaces of two layers, and interfacial fragmentation or peeling occurs under slight external peeling forces[43]. ATI, on the other hand, exhibits distinct interfacial toughness, which may be attributed to in situ layer-by-layer photopolymerization that allows physical entanglement and chemical cross-linking between polymer networks at the interface between ATI-BN and ATI-BN to achieve strong interfacial bonding (Fig. 2f, Supplementary Fig. 16)[44,45]. To validate the impact of the asymmetric structure on the viscoelastic behavior of the ionogels, a comprehensive rheological analysis of ATI was performed. The results show that the storage modulus (G′) of ATI is larger than the loss modulus (G″) in a wide range of amplitude and frequency sweepings as well as temperature ramping (Fig. 2g). The above results indicate that the polymer network inside the ATI exhibits notable dynamic reversibility under complex external forces, and exhibits an intrinsic gel state over a wide range of operating temperatures, and the asymmetric two-phase interface does not fail due to the occurrence of fracture.

## Reversible thermochromic response triggered by dynamic phase separation

The thermochromic behavior of ionogels originates from dynamic phase separation within ATI-B. At low temperatures, the IL remains uniformly dispersed in the PBA matrix via ion-dipole interactions (primarily cations and the -C = O group in ester moieties), maintaining a homogeneous single phase (Fig. 3a)[46]. With increasing temperature above the LCST, uniformly distributed anions and cations aggregate around the polymer network[47], forming distinct liquid domains that result in a two-phase system (Fig. 3b, Supplementary Fig. 17 and Movie 3). When dynamic phase separation occurs, a large number of "island-like" IL domains inside ATI-B scatter light, resulting in an opaque state (Fig. 3c). The transmittance of ATI-B in the visible light range decreases from more than 85% to less than 10% when the temperature increases from 20°C to 45°C, and haze within the visible light spectrum increases dramatically from below 10% to around 94% (Supplementary Figs. 18 and 19). In contrast, ATI-BN exhibits thermal stability within the same temperature range (Supplementary Figs. 20 and 21). The LCST of ATI-B was analyzed by the differential scanning calorimetry (DSC) technique, and a small enthalpy change was found around 34°C, indicating that the material undergoes a phase transition (Supplementary Fig. 22 and Note 2). The lower $\Delta H$ implies that the ionogel has considerable dimensional stability, since a large enthalpy change always implies a large-scale phase separation that occurs in parallel with the liquid efflux[48]. By varying the ratio of the polymer to the IL, the LCST of ATI-B can be tailored between 28°C and 41°C to meet specific

requirements (Supplementary Fig. 23 and Table 2). Except for LCST, the formulation adjustments to the ATI-B will not significantly affect the mechanical properties or low-temperature transparency of ATIs (Supplementary Fig. 24 and Table 3).

For LCST-type TCPs, the sensitivity and long-term resistance to temperature changes directly affect their performance in practical application[49]. ATI exhibits rapid response and significant optical changes above and below LCST (Fig. 3d). It is noteworthy that, owing to the intrinsic nonvolatile nature and amorphous structure of the ILs, along with the suppression of IL crystallization through ion–dipole interactions[50], ATI remains transparent without undergoing transparency transition caused by ice crystal formation, even after prolonged exposure to a low temperature of −70 °C (Supplementary Fig. 25, Note 3, and Movie 4). Even after prolonged exposure to low temperatures, ATI undergoes opacity switching in a short period when returning to temperatures above LCST (Supplementary Fig. 26). In contrast, the growth of ice crystals in PNIPAM hydrogels leads to the destruction of the integrity of the polymer network after multiple freeze-thaw cycles (Supplementary Fig. 27). In 100 cycles of testing from −45°C to 40 °C, the transmittance of ATI to 550 nm wavelength light does not undergo significant degradation (Supplementary Fig. 28). Moreover, ATI does not experience significant dimensional shrinkage or functional failure even when subjected to long-term operating conditions above LCST (Supplementary Fig. 29). The above advantages enable ATI to effectively circumvent the limitations of low-temperature freezing and high-temperature water loss faced by hydrogel-based TCPs. The stability of ATI was further evaluated under a series of complex accelerated aging protocols, including cyclic high-temperature/humidity conditions, salt spray tests, and UVA/UVB radiation. The experimental results confirm its robustness (Supplementary Figs. 30 and 31). It is noteworthy that while intense UV exposure leads to some photo-induced yellowing, this phenomenon is temporary. The discoloration significantly recedes after storage in ambient air, and the thermochromic functionality and low-temperature tolerance remain almost unaffected (Supplementary Fig. 32).

The microscopic interactions in thermosensitive dynamic phase separation were analyzed by in situ characterization and computational chemistry techniques. The temperature-dependent FTIR results (Fig. 3e) indicate a redshift of the characteristic peaks of ATI-B (from 1052 to 1050 cm$^{-1}$ and 1348 to 1346 cm$^{-1}$), suggesting a weakening of the ion–dipole interaction between the IL and polymer[51]. This observation is strongly corroborated by density functional theory (DFT) calculation, which shows a significant decrease in the PBA-ion interaction energy per unit area (Supplementary Fig. 33). Likewise, the temperature-dependent Raman spectroscopy (TDRS) reveals a shift of the $v$(-N-S) peak (Fig. 3f, Supplementary Fig. 34) to lower wavenumbers, implying enhanced Coulomb interaction between [Emim]$^+$ and [TFSI]$^-$[52]. This phenomenon is perfectly reflected in the molecular dynamics (MD) simulation, where the [Emim]$^+$-[TFSI]$^-$ interaction energy increases with rising temperature and becomes dominant (Fig. 3g). Furthermore, two-dimensional correlation infrared spectroscopy (2DCOS FTIR) analysis suggests that the phase separation is triggered by the preferential dissociation of anions from the polymer, followed by cation re-association (Supplementary Fig. 35, Table 4 and Note 4). Taken together, these tightly coupled experimental and theoretical results provide compelling evidence for an entropy-driven dynamic phase separation mechanism above the LCST, wherein enhanced Coulomb forces promote the aggregation of solvent molecules detached from the polymer (Supplementary Note 5)[53].

## Tough mechanical properties enabled by in situ phase separation

The in-situ phase-separated structure of ATI-BN features a bicontinuous network comprising a polymer-rich phase (dominated by rigid

NIPAM segments with strong intermolecular hydrogen bonding) and a solvent-rich phase (dominated by elastic, gel-like BA segments). This structure enables an interphase load-transfer mechanism and energy-dissipation pathway[54], ultimately toughening the ionogel (Supplementary Fig. 36). Under stress, the rigid polymer-rich phase preferentially dissipates energy, while the solvent-rich phase acts as a "hidden length" that accommodates stress through large deformation. The toughness, Young's modulus, and elongation at break of ATI increase by 2500%, 3390%, and 110%, respectively, compared to the homogeneous pure PBA ionogel (Fig. 4a, b, Supplementary Fig. 37). However, the overall mechanical properties of ATI are relatively weak compared with those of pure P(BA-co-NIPAM) ionogel (Supplementary Fig. 38), which is consistent with the toughening law of in situ phase separation strategy[55]. When compared with pure PBA ionogel and PNIPAM hydrogel, ATI demonstrates significant advantages in terms of practical mechanical performance (Fig. 4c, d, Supplementary Fig. 39 and Movie 5). The interfacial strength between ATI-B and ATI-BN was tested by lap-shear experiments, and the interfacial shear strength of ATI reaches 600 kPa, which is 110 times higher than that of the physical bilayer ionogel; and the strong interfacial strength of ATI shows a significant advantage even when compared with commercial adhesive tapes (Fig. 4e, Supplementary Fig. 40). We simulate the interfacial peeling process in physical bilayer ionogel and ATI using finite element analysis (FEA). The results show that under pulling of vertically upward force, ATI exhibits higher stress concentration than the bilayer ionogel, and the near-perpendicular angle between upper and lower layer indicates that the interfacial bonding of ATI is significantly stronger (Fig. 4f). After puncture resistance testing, ATI maintains structural integrity without observable morphological damage or interfacial failure, demonstrating high reliability in complex environments (Fig. 4g). During cyclic compression tests with increasing strain, ATI consistently displays complete hysteresis loops in stress-strain curves and retains structural integrity throughout (Fig. 4h, i). Performing more uninterrupted compression cycles (70% compression) on ATI, it is found that, except for the first cycle, the hysteresis loops of all cycles almost overlap, indicating that the elastic polymer network in ATI has a highly reversible deformation capacity and fatigue resistance (Fig. 4h-ii)[56]. Robust mechanical properties and a suitable LCST range are important indicators for evaluating the performance of TCPs in practical applications, especially in smart windows and flexible optical displays. Comparing with the reported TCPs, ATI has significant advantages in mechanical performance (in terms of tensile strength) (Fig. 4i)[57–61]. The transition temperature ($\tau_c$) is also within the range of the average high temperatures in the human living environment (30–40°C), ensuring thermal activation under ambient conditions. Notably, the property of not requiring additional encapsulation allows ATI to be directly applied to non-planar objects and stably maintain the thermochromic function (Fig. 4j). The realization of the above advantages is expected to bridge the gap between rigid TCP devices and flexible smart systems.

## Smart cooling window enabled by passive thermochromic optical switching

Passive thermochromic smart windows dynamically adjust UV-Vis-NIR light transmittance in response to temperature changes (Fig. 5a), thereby enabling building energy savings[62]. Unlike conventional TCP-based smart windows requiring glass encapsulation, ATI smart windows are fabricated by direct application of ionogel onto window surfaces. Rich hydrophobic groups on ATI (e.g., butyl chains from BA and perfluoroalkyl chains from IL) impart intrinsic hydrophobicity[63], achieving water contact angles up to $135 \pm 1°$—significantly higher than tempered glass and PNIPAM hydrogel (Fig. 5b-i, Supplementary Fig. 41). The surface adhesion work of ATI-BN is calculated to be 20.8 mN m$^{-1}$, which is about 510% lower compared to tempered glass, indicating that water droplets have lower sliding resistance on the ATI

surface (Fig. 5b-ii). Further field tests demonstrate that water droplets easily slipped off the ATI surface, and the surface could be easily cleaned with water directly (Fig. 5b-iii, Supplementary Movie 6). On the other hand, the abundance of polar groups on the surface provides ATI with high adhesion ability, enabling it to autonomously adhere to the surfaces of various transparent building materials (Fig. 5c), including tempered glass, polymethylmethacrylate (PMMA), polyvinyl chloride (PVC), and polycarbonate (PC). In particular, 90° peel strength of ATI on glass is greater than 400 N m$^{-1}$ (Supplementary Fig. 42). The above advantages ensure the convenience and practicality of ATI when used as a smart window.

For smart optical modulation and cooling performance, the transmittance of ATI to UV-Vis-NIR light waves is negatively correlated with temperature (Fig. 5d). At 20 °C, ATI is highly transparent with a solar transmittance ($T_{sol}$) of 75.7%. When warmed up to 45°C, ATI undergoes a transparency transition, and $T_{sol}$ decreases to 18.9%, showing a notable $\Delta T_{sol}$ of 56.8%. Similarly, the luminous transmittance ($T_{lum}$) of ATI shows a $\Delta T_{lum}$ of up to 78.3% (Supplementary Table 5 and Note 6). Compared to the recently reported adjustable phase-change materials for energy-saving smart windows, ATI still demonstrates competitive solar modulation capabilities (Supplementary Fig. 43). Under simulated sunlight (xenon lamp), the model house equipped with ATI smart window exhibits a slower heating rate compared to the tempered glass reference house. After heating termination, the house with the ATI window shows a slower cooling rate, demonstrating superior thermal insulation performance compared to conventional windows. (Fig. 5e). The performance of the ATI-based smart windows in regulating natural sunlight real daytime conditions was further evaluated (Fig. 5f). When the ambient temperature reaches about 40°C at midday in summer, the room temperature of the model house with air windows (semi-open system) reaches 52°C; with glass windows (fully closed system) reaches an intimidating 66°C; and with ATI smart windows (fully closed system) is only 56°C (Fig. 5g). To assess the energy-saving potential of ATI smart windows, annual energy simulations were performed using EnergyPlus on a building model equipped with ATI windows (Supplementary Fig. 44). Compared to clear windows with pure glasses, ATI smart windows deliver more significant energy savings in China's warmer urban areas, highlighting their potential for application in solar-regulated devices (Fig. 5h). For the aesthetic and personalized design of windows, ATI can be easily customized into complex window patterns or colors through simple reprocessing such as templating, cutting, dyeing, and modular splicing, without affecting its intrinsic thermally-induced transparency transition (Supplementary Fig. 45).

## Active dynamic optical display enabled by indirect/direct Joule heating

The optical properties of ATI vary with ambient temperature. Leveraging the intrinsic advantages of temperature as a stimulus—ubiquitous availability, non-contact operation[64], and precise controllability—establishes ATI as a promising active optical switch for temperature-controlled applications (Fig. 6a). Integrating a flexible conductive heating layer as a thermal switch, this indirect Joule heating approach enables reversible optical modulation between transparent and opaque states (Fig. 6b). By customizing the conductive heating layer with patterning, the same opaque patterns can be displayed on ATI, thus achieving the effect of dynamic localized optical display (Fig. 6c-i). Since the conductive heating layer employs a flexible graphene heating sheet driven by a direct current (DC) power supply, the ATI-based optical display device is no longer limited by a rigid plane substrate and can be applied to curved surface carriers (Fig. 6c-ii). When this localized optical display device is applied to a window, dynamic information display can be achieved. Indoor users can display information externally on the window utilizing temperature stimulation,

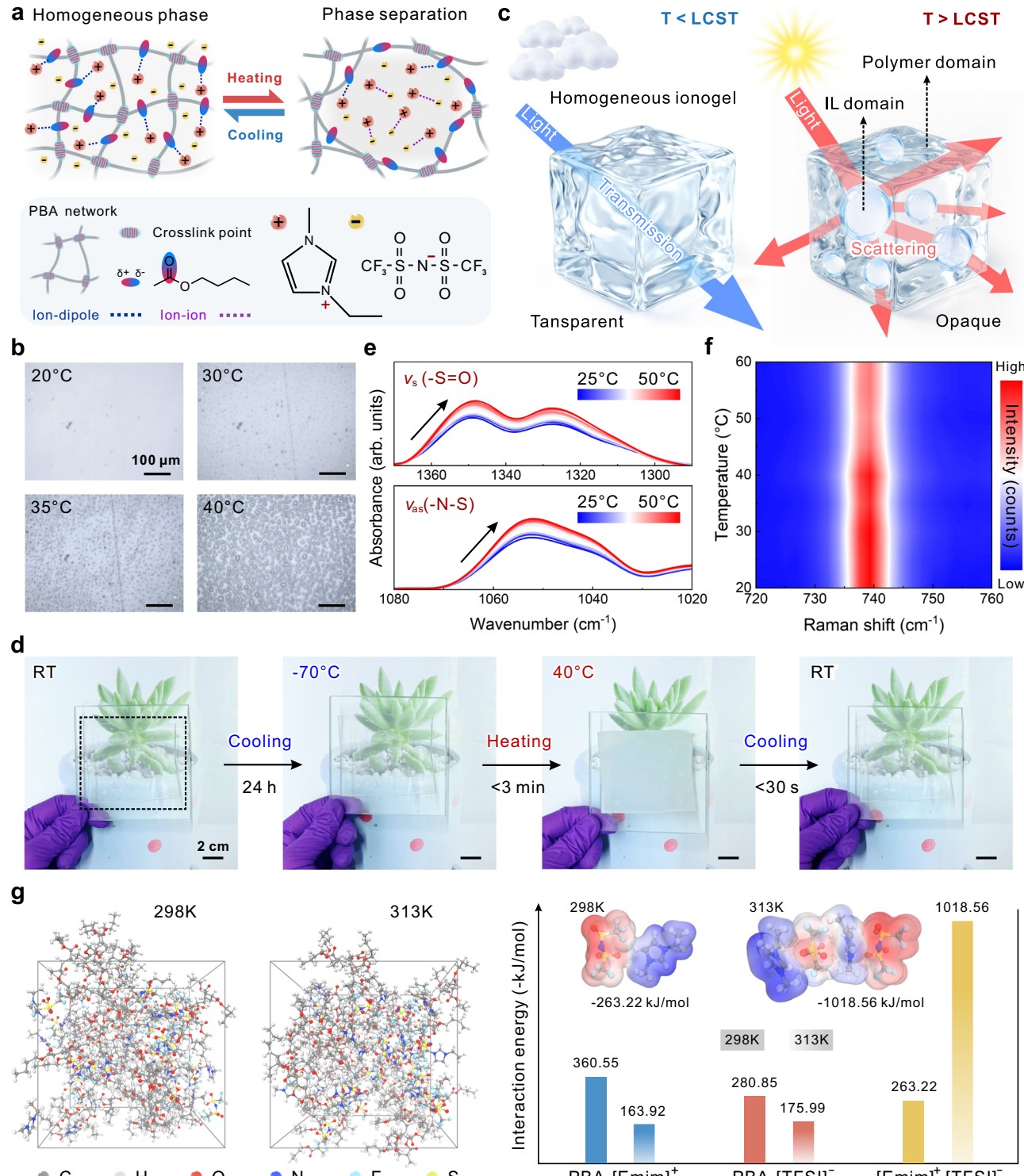

**Fig. 3 | Dynamic phase separation principle and microscopic interaction analysis of ATIs. a** Proposed schematic of the thermosensitive dynamic phase separation in ATI-B. **b** Optical microscopy images of ATI during the heating process. **c** Proposed schematic of the propagation path of light inside ATI-B below (left) and above (right) the LCST. **d** Optical photos of ATI under different temperatures. **e** Temperature-dependent FTIR spectra. **f** Temperature-dependent Raman spectra (color scale: red, high intensity; blue, low intensity). **g** MD snapshots of PBA and [Emim][TFSI] at 298 K and 313 K. The calculated interaction energies of ion-dipole interactions (PBA-[Emim]⁺, PBA-[TFSI]⁻) and ion-ion interactions ([Emim]⁺-[TFSI]⁻) in ATI at 298 K and 313 K; insets show the ESP of the optimized structure of [Emim]⁺-[TFSI]⁻ at the two temperatures (color scale: red, negative; blue, positive).

such as an emergency distress signal (Supplementary Fig. 46). Such innovative attempts allow future smart windows to serve not only the energy efficiency of a passive-cooling building, but also have the potential to be applied as a dynamic optical display.

To further demonstrate the potential application of ATI in the next generation of optical display systems, we combine ATI's Joule heating with its optical properties to realize integrated global dynamic optical switching. This behavior leverages the inherent ionic

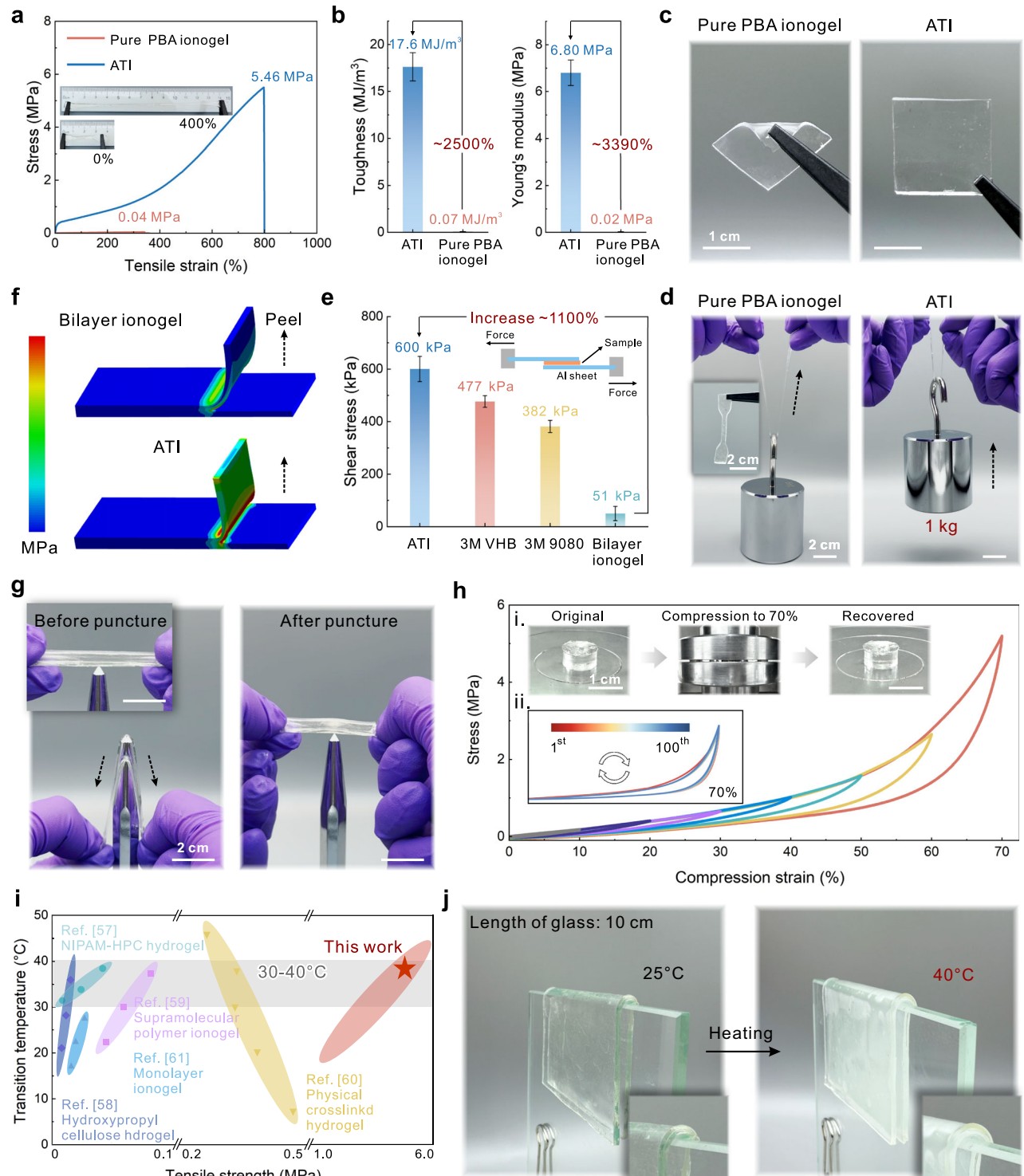

**Fig. 4 | Tough mechanical properties of ATI.** Comparison of the mechanical properties of ATI and pure PBA ionogel. Including **a** Stress-strain curve; **b** toughness and Young's modulus, error bars represent standard deviation (n = 3); **c** optical photograph. **d** Comparison of load capacity in real-world scenarios, ATI can easily lift 1 kg of weight, which is not possible for pure PBA ionogel. **e** Comparison of the interfacial shear stress of ATI with other adhesive materials, including commercially available 3 M VHB, 3 M 9080, and bilayer ionogel, error bars represent standard deviation (n = 3). The inset shows a schematic of the lap-shear experimental setup for measuring interfacial strength. **f** FEA mechanical simulation of bilayer ionogel and ATI, where the darker color (red) is subjected to higher stress. **g** Puncture resistance performance of ATI. The appearance of ATI before (inset), during (left), and after (right) puncture is not significantly damaged. **h** Stress-strain curves of ATI under different compression cycles; insets show optical photographs before, during, and after compression, as well as stress-strain curves under 100 compression cycles. **i** Comparison of the properties of ATI with reported thermochromic materials, including tensile strength and effective LCST range. **j** Photographs of ATI being used on a shaped surface at 25℃ (left) and 40 ℃ (right).

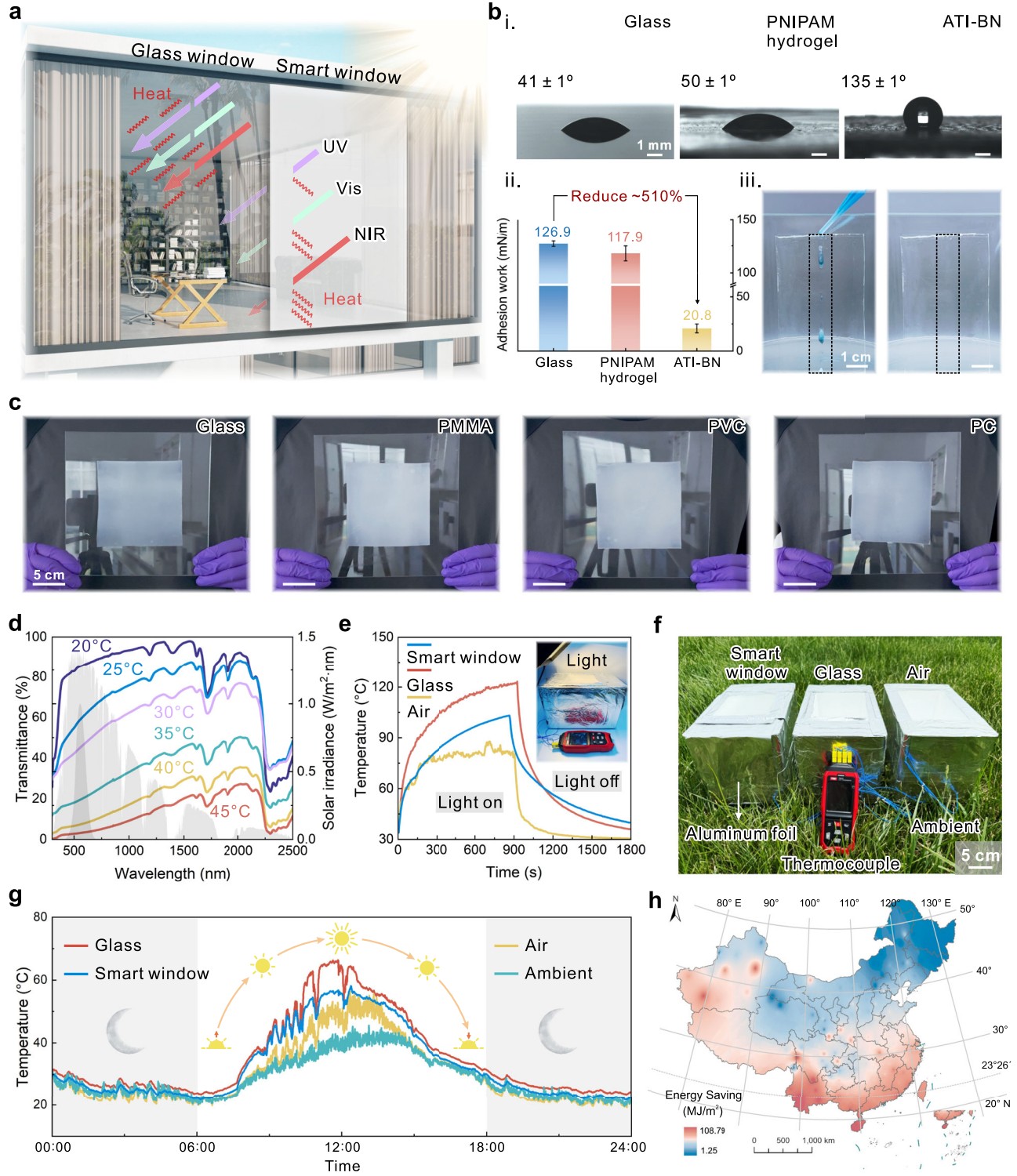

**Fig. 5 | Passive smart cooling windows realized by ATI. a** Proposed light modulation schematic for ATI smart windows. **b** Hydrophobicity of ATI smart windows, including contact angle (i) and adhesion (ii) of ATI-BN compared with glass and PNIPAM hydrogel, as well as simulated raindrop sliding test (iii). Error bars represent standard deviation ($n = 3$). **c** Demonstration of autonomous adhesion of ATIs on different common transparent substrates. **d** UV-Vis-NIR transmittance spectra of ATI-based windows at 20–45°C, together with the solar irradiance intensity (air mass 1.5 global) and photopic luminous efficiency. **e** The temperature variation of model rooms with different windows under the illumination of simulated sunlight (xenon lamp) over time; inset is an optical photograph of the test site. **f** On-site optical photos of the model rooms during the cooling performance testing process. **g** Variation of room temperature in model rooms with different windows within 24 h (Xi'an, 34°15′ N, 108°40′ E, May 20, 2025). **h** China energy-saving map of the building model with ATI smart windows.

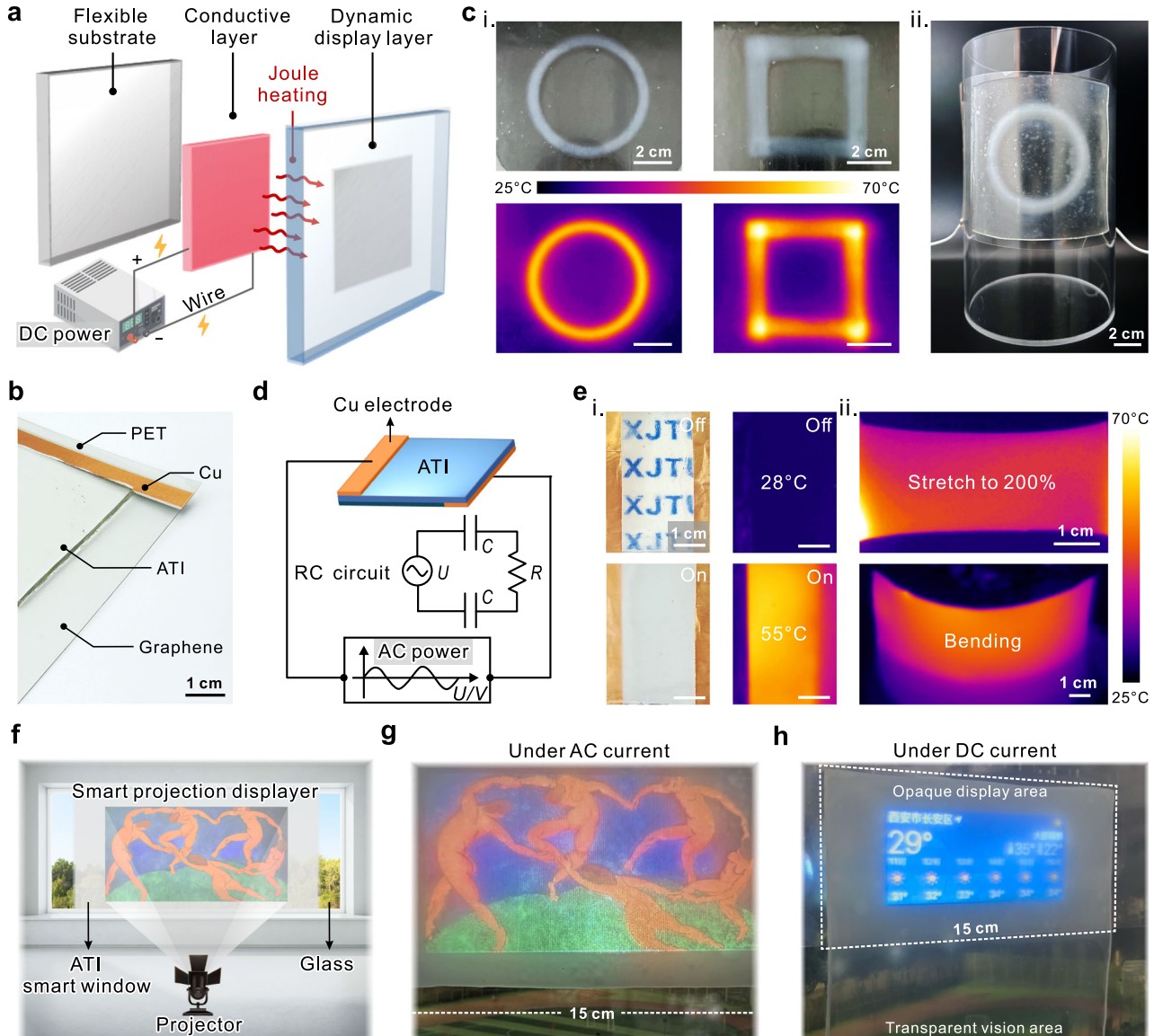

**Fig. 6 | Flexible dynamic optical display realized by ATI. a** Schematic of a flexible dynamic display device driven by localized indirect heating. **b** Optical photographs of a localized optical display device. **c** Demonstration of localized optical displays, including optical photographs and thermal infrared images in the planar state (i), and localized optical displays on curved substrates (ii). The DC voltage for the conductive Joule heating layer is 24 V. **d** Schematic and equivalent circuit diagram of the ATI-based global Joule heating system. **e** Demonstration of in situ Joule heating of ATI using an AC power supply (180 V, 10 kHz), including optical and thermal infrared images before and after power on in the planar state (i), as well as thermal infrared images of ATI in the stretched and bent states when power is on (ii). **f** Schematic of a smart projection display system based on ATI optical display switching; and **g** Realistic demonstration of the ATI smart projection display system, with the famous painting "Dance" (Henri Matisse, 1910, public domain work) projected globally. **h** Demonstration of a dynamic localized projection display. The upper half is an opaque area due to localized heating, used to display the day's weather conditions, and the lower half is highly transparent because it is not heated.

conductivity of ionogels, enabling Joule self-heating through ionic conduction[65]. ATI exhibits temperature-dependent ionic conductivity, while maintaining high robustness under thermal cycling conditions (Supplementary Fig. 47). As a proof-of-concept demonstration, a high-frequency alternating current was applied to the ATI here using two Cu electrodes and an alternating current (AC) power supply (Fig. 6d). Within 15 s after applying voltage (180 V, 10 kHz), the local temperature of the ATI reaches up to 55℃ and undergoes a complete transparent-to-opaque transition (Fig. 6e-i, Supplementary Movie 7). Further testing indicates that the temperatures resulting from Joule heating on the surface of ATI are positively correlated with the applied AC voltage (Supplementary Fig. 48). This global photoswitch display maintains effective heating effect and stable light scattering ability

even when subjected to stretching and bending (Fig. 6e-ii). On this basis, a global dynamic light switch based on in situ Joule heating is designed and demonstrated as a kind of intelligent projection screen (Fig. 6f). ATI with reduced light transmittance acts as a natural projection screen at room temperatures above the LCST. When the ambient temperature is below the LCST of ATI, a transparent-to-opaque transition can also be achieved through in situ Joule heating. Projections of classical artworks onto the opaque ATI surface maintained high color saturation and display resolution (Fig. 6g). Finally, a smart screen for localized projection by indirect Joule heating is also demonstrated, and localized weather information can be displayed on a smart window without affecting the perspective effect (Fig. 6h, Supplementary Movie 8). The above attempts not only offer innovative

ideas for next-generation optical display devices but also address the shortcomings of existing smart windows, which have only a single function.

## Discussion

This study developed Janus structural asymmetric thermochromic ionogels (ATIs) via synergistic dynamic in situ phase separation. The ATIs combine a thermoresponsive PBA/IL layer for optical switching with a toughened P(BA-co-NIPAM)/IL layer for mechanical support, achieving an optimal LCST (34 °C), low temperature resistance (retaining transparency down to −70 °C), and high robustness (tensile strength >5 MPa, interfacial shear strength >600 kPa). The dynamic phase separation mechanism, driven by entropy-mediated ion aggregation above the LCST, enables rapid and reversible opacity switching ($\Delta T_{lum} = 78.3\%$) over 100 cycles. Unlike conventional TCMs, ATIs eliminate encapsulation requirements and maintain high environmental stability through intrinsic hydrophobicity (contact angle >135°), non-volatility, and strong adhesion to diverse substrates. As passive smart windows, they reduce solar heat gain by 56.8%, lowering indoor temperatures by 10 °C compared to glass. For active optical control, patterned indirect Joule heating enables localized displays on curved surfaces, while global Joule self-heating (180 V AC) through ionic conduction creates deformable projection screens. This work is expected to bridge rigid TCMs and flexible systems, offering a scalable platform for adaptive optics in energy-efficient buildings and interactive displays. To enhance the application potential of ATI in future, the following two aspects should be further investigated: i) optimizing the forming technology of asymmetric structures to mitigate constraints imposed by mold area and intricate preparation processes on the large-scale production of ATI; and ii) leveraging the modifiability of polymers to continuously improve the environmental robustness of ATI, particularly under complex extreme conditions such as durability at ultra-high temperatures (exceeding 100 °C), toughness at ultra-low temperatures (below liquid nitrogen temperature), and performance in multifaceted harsh environments (e.g., under strong UV irradiation coupled with high temperature and humidity).

## Methods

### Preparation of asymmetric thermochromic ionogels

Asymmetric ionogels were prepared via a sequential UV-curing process. Firstly, 23.4 mmol of BA was ultrasonically dissolved in 5.1 mmol of [Emim][TFSI]. Irgacure 1173 (0.05 mol% relative to monomer) and the cross-linker ethylene glycol dimethacrylate (EGDMA, 0.05 mol% relative to monomer) were added to form the ATI-B precursor solution. After nitrogen purging to remove dissolved oxygen, this solution was injected into a sealed glass mold (1 mm silicone spacer) and cured under UV light (365 nm, 24 W) for 60 s. Subsequently, the mold was disassembled, and the spacer was replaced with a 2 mm one. The ATI-BN precursor solution was prepared by ultrasonically dissolving 15.6 mmol of BA and 17.6 mmol of NIPAM in 2.5 mmol of [Emim][TFSI], with Irgacure 1173 and EGDMA added at the same 0.05 mol% (relative to monomer). Following dissolved oxygen removal, this solution was injected into the mold and subjected to in situ UV-curing (365 nm, 48 W). Due to significant exothermic heat release during NIPAM radical polymerization in [Emim][TFSI], this curing step was performed in an ice-water bath.

### Material characterization

FTIR (Nicolet iS 50, Thermo Fisher Scientific, USA) was used for characteristic peak measurements with a resolution of 0.4 cm⁻¹ and a wavelength range of 4000-800 cm⁻¹. Temperature-dependent diffuse reflectance FTIR tests were performed in the range of 25–50 °C utilizing a temperature-controlled accessory. The temperature-dependent FTIR data were processed by the 2D Correlation Spectroscopy analysis v1.20 plug-in in Origin. The surface morphology of the cross sections of ATIs

was observed using an optical microscope (Axio Scope A1, Zeiss, Germany) and a Field emission scanning electron microscope (FE-SEM, GeminiSEM 500, Zeiss, Germany), which was brittlely fractured using liquid nitrogen before the FE-SEM observation. Energy dispersive X-ray spectroscopy (EDS) analysis was performed simultaneously with FE-SEM. Raman spectroscopy and spatial Raman mapping of ATIs were performed using a Raman imaging microscope (InVia Qontor, Renishaw, UK). The excitation laser wavelength was 532 nm, and a spatial resolution of 2 μm was used. Dynamic viscoelasticity tests were performed on ATIs using a rheometer (MCR302, Anton Paar, Austria), including amplitude-, frequency-, and temperature-sweeping tests. Small-angle X-ray scattering (SAXS) and wide-angle X-ray scattering (WAXS) experiments were conducted with an Xeuss 3.0 system (Xenocs, France). X-Ray wavelength: 1.54189 Å; detector: Eiger2R 1 M; sample to detector distance: 1000 mm; WAXS experimental environment: vacuum, 20 °C, 30 °C, and 40 °C. The lengths of the scattering vector q were 0.02 Å⁻¹ to 0.25 Å⁻¹ for SAXS, and were 0.64 Å⁻¹ to 3.20 Å⁻¹ for WAXS. Perform fitting calculations on SAXS results using SASfit software (version 0.94.12) and Origin software (version 9.9.0.225). The phase separation size in ATI-BN was characterized using the Debye-Anderson-Brumberger (DAB) model. The appropriateness of this model for determining correlation lengths in random, non-granular two-phase systems is confirmed by its assumption of smooth interfaces, a feature consistent with the bicontinuous structure of ATI-BN. The pair correlation function γ(r) is given by Eq. 1:

$$\gamma(r) = \exp(-r/\xi) \tag{1}$$

where r is radial distance.

The macroscopic scattering cross-section in the DBA model is given by Eq. 2:

$$I(q) = \frac{(\pi\Delta\eta\xi^3)^2}{(1+q^2\xi^2)^2} \tag{2}$$

where q is scattering vector; I(q) is scattering intensity; ξ is correlation length; Δη is scattering length density contrast.

Bragg spacing in WAXS is given by Eq. 3:

$$d = \frac{2\pi}{q} \tag{3}$$

where d is the real-space feature distance; q is the scattering vector, defined by Eq. 4:

$$q = \frac{4\pi\sin\theta}{\lambda} \tag{4}$$

where θ is the half-scattering angle; λ is the wavelength of the incident X-rays.

### Thermochromic performance and low-temperature resistance test

Thermosensitive dynamic phase separation of ATIs was observed using an optical microscope (BX51, Olympus, Japan) with a temperature-controlled accessory. Measurement of glass transition temperature and critical transition temperature of ionogels using a differential scanning calorimeter (DSC2500, TA Instruments, USA). The transmittance and haze values of ATI in the visible range (380-780nm) were measured using a spectrometer (QEPro, Ocean Optics, USA) with a temperature-controlled accessory over 20–45 °C. The haze

value (*H*) can be calculated using the following Eqs. 5 and 6:

$$T_{\text{diffusive}} = T_{\text{total}} - T_{\text{specular}} \qquad (5)$$

$$H = \frac{T_{\text{diffusive}}}{T_{\text{total}}} \times 100\% \qquad (6)$$

where $T_{\text{total}}$ is the total transmitted luminous flux; $T_{\text{diffusive}}$ is the diffusive transmitted luminous flux. $T_{\text{specular}}$ is the specular transmitted luminous flux[66]. $T_{\text{total}}$ is measured using an integrating sphere; $T_{\text{specular}}$ is obtained through direct vertical measurement.

Different temperature environments were created using blast ovens or hot water (40 °C), room temperature water (RT 25°C), dry ice alcohol baths (−70 °C), and liquid nitrogen (−196°C) to test the thermochromic properties of ATIs and the ability to maintain transparency at low temperatures. For thermochromic stability testing, the transmittance at 550 nm was read after 2 min of holding at each temperature point.

## Data availability
The data that supports the findings of the study are included in the main text and supplementary information files, or from the corresponding author upon request. Source data are provided with this paper.

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

## Acknowledgements

We acknowledge funding of the National Natural Science Foundation of China (62475212 and 61975162, X.L.), the Space Application System of China Manned Space Program (No. KJZ-YY-NCL10, S.Y.), the National Key R&D Program of China (2022YFE0109500, S.Y.), and the Shaanxi Province Natural Science Foundation (No. 2020JM-062, X.L.).

## Author contributions

G.D. and X.L. conceived the concept and idea. The entire experimental procedure was designed by G.D., with assistance from J.L. and C.W., and the three of them jointly coordinated the project. G.D. and J.L. designed the materials and conducted experiments. G.D., C.W., J.L., Y.N., and Y.Y. characterized, and analyzed the materials. G.D., J.L., and Y.X. conducted field testing of materials and construction of heating devices. All authors also discussed and interpreted the experimental data. G.D. performed the theoretical calculations and energy consumption simulations. G.D. wrote the manuscript with contributions from all authors. X.L. and S.Y. supervised the project and reviewed the paper. All authors discussed the results and commented on the paper.

## Competing interests

The authors declare no competing interests.
