## [Transparent Peer Review file · Nature Communications]

Tough asymmetric thermochromic ionogels via dynamic in situ phase separation for dual-modal smart optical switching

Corresponding Author: Professor Xuegang Lu

Version 0:

Reviewer comments:

Reviewer #1

(Remarks to the Author)

This manuscript reports on the fabrication of an asymmetric thermochromic ionogel with a bilayer structure comprised of a thermoresponsive light-scattering layer and a tough supporting layer. The thermochromic layer serves as the active layer to modulate light transmittance upon temperature changes. The main thermochromic mechanism lies in the thermoresponsive phase separation of IL and PBA at temperatures higher than its LCST. Meanwhile, the microphase-separated structures of P(BA-co-NIPAM) serve as energy dissipation mechanism to yield a tough ionogel. The bilayer can shift from transparent to opaque states as the temperature shuttles between temperatures below and above the LCST. The thermochromic property is utilized to modulate solar radiation through windows, which is demonstrated to lower the room temperature. The ion conductive ionogel shows Joule heating characteristic, which is used to motivate phase transition through electro-stimulation. This research is recognized as an interesting contribution to the field of smart materials with switchable and tunable optical properties. Although the demonstration to potential applications is fascinating, the fundamental sections on the materials design, synthesis, and characterizations are not clear or adequate to establish the new concept. Besides, the biomimetic design principle is not correct or necessary. The manuscript needs extensive revisions before it could be considered for publication in Nature Communications.

- 1, The biomimetic concept is not correct or necessary. The asymmetric bilayer structure has nothing to do with the beetle carapace. No resemblance could be found between the current bilayer ionogel and the composite bilayer structure of the carapace. Such an analogue in fact harms the main idea of this study.
- 2, Some important terms, including phase separation and dynamic phase separation, are used without a clearly and precise definition. It is difficult to understand them in the ionogels and the thermochromic behavior.
- 3, LCST is a most important characteristic of the ATIs (particularly for ATI-B). But this LCST is not clearly defined or explained. What is the mechanism? How to determine LCST? The DSC curve shows two transitions and the author assign the one at 34C as the LCST. What is the criterion? Supplementary Figure 17 shows different transition temperatures of ATI-Bs with different formulations. But why should the author identify them as LCST? How to tune the LCST (if it is true)? This section is very important for the whole study.
- 4, Concerning the SAXS and WAXS results, the authors attribute the scattering peaks to the contraction of the ionic clusters and the increase of pi-pi stacking distance caused by enhanced ion-ion interactions. This is quite arbitrary. It is not clear why should the ion-ion interaction be enhanced. What are the so-called ionic clusters? The authors claim the formation of bicontinuous structures in the ATI-BNs. But the SAXS pattern does not provide any useful information on the phase separated structures of ATI-BNs.
- 5, The interface robustness is key to the bilayer structure. The interface strength should be tested.
- 6, Most schematic illustrations look fancy. But some of them are over rendering. In figure 3a, for example, what is the ion-dipole interaction? What is the dipole? In Figure 3b, it is difficult to image that the light can be "scattered" throughout the opaque ionogel. It is violating the basic optical physics.
- 7, It is very important to separately characterize the optical properties and transitions of ATI-Bs and ATI-BNs. So far, no picture or systematic data are presented for these single layers. In most cases, photos of ATIs are presented, so it is difficult to identify the contribution to the optical transition.
- 8, The solar modulation capability is low even at the visible range. This reminds of Figure 1g, where the authors compare the performances of the current TCMs with those reported in the literature. Unfortunately, the thermochromic and solar modulation performance is not included in this radar chart for some reason. It is tricky and unfair to mislead the readers.
- 9, Figures 5h and 5i do not add much to the manuscript.
- 10, For the Joule heating, what is the voltage used to heat the ionogel above its transition temperature? "Equivalent opacity

can be achieved below the LCST via in situ Joule heating." What does this mean? Is the temperature of the ionogel below its LCST when it remains opaque? This statement is quite confusing and misleading. What is the merit or significance to use such an opaque "screen" for projection?

Reviewer #2

(Remarks to the Author)

This manuscript reports an asymmetric thermochromic ionogel (ATI) inspired by Janus structures: a thermally induced light-scattering layer (tunable LCST) is integrally coupled to a mechanically tough support layer via sequential radical polymerizations and in-situ phase separation. The device claims high optical contrast, low-temperature tolerance, high toughness, self-adhesion without edge sealing, and both passive and active optical switching, with demonstrations spanning model-building windows and patterned heating for simple displays. The thermal scattering and high toughness are considered through the covalent/entanglement interface of "dynamic thermally induced phase separation (scattering layer) in-situ phase separation (toughening layer)", which is rarely seen in PNIPAm hydrogel or single-phase ionic gel systems. The concept is compelling. To meet Nature Communications standards, the manuscript would benefit from stronger evidence connecting thermal, optical, and mechanical responses; from durability and lifetime statistics; and from explicit safety and energy metrics for device operation.

1. Given the ionogel's advertised customizability, I encourage the authors to expand the property space they present so that readers can see how formulation choices map onto performance.
2. Because the switching mechanism is fundamentally scattering-based, haze is the pivotal figure of merit for privacy glazing and projection/screen-like use. Please report haze as a function of wavelength and temperature across the visible range and, ideally, include angle-resolved measurements.
3. How does the polymer ionogel system maintain a transparent phase below the LCST at deep sub-zero temperatures. To what extent is the observed stability a general property of ionic liquids, and can the authors compare hydrophobic, low-water-activity ILs with more hygroscopic ones to establish whether hydrophobicity and low volatility are necessary conditions for low-temperature durability and frost-free transparency?
4. The device reads as mechanically robust and therefore promising; nevertheless, optical components must withstand harsh environments and strong UV exposure in real service. I recommend adding accelerated aging test like high temperature-humidity, UV-A/B exposure, or salt-fog/corrosive atmospheres.
5. To illustrate practical impact, the measured optical states should be coupled to building-scale energy simulations under representative climates.
6. The current SAXS analysis is qualitative; please perform quantitative structure-factor fitting to extract real-space correlation lengths or domain sizes and present their temperature-dependent evolution alongside scattering intensity.
7. please undertake a careful language and figure audit. Avoid typos like, in Fig. 5d the axis label should be labeled "wavelength."

Reviewer #3

(Remarks to the Author)

The manuscript by Du et al. presents a technically impressive and well-executed study on a bio-inspired, asymmetric thermochromic ionogel (ATI). The authors have successfully engineered a material with an outstanding combination of mechanical toughness, environmental resilience (especially low-temperature performance), and dual-mode optical switching capabilities. The experimental work is thorough, spanning multi-scale characterization and theoretical simulations. Overall, this work is highly innovative and significant, and its findings will be of broad interest to the fields of smart materials, flexible electronics, and energy-saving buildings. I recommend that this manuscript be accepted for publication in Nature Communications after the authors address the following minor points.

Specific Comments for Revision:

1. In the main text (Fig. 3), the authors use MD simulations to show that an increase in temperature weakens the interaction energy between PBA and the ions while significantly strengthening the ion-ion interactions. This is a central argument. It is suggested that this point be more explicitly linked in the main text to the peak shifts observed in the FTIR and Raman spectra. This would create a stronger connection between theory and experiment and more directly lead to the conclusion of "entropy-driven phase separation." For example: "The temperature-dependent FTIR results (Fig. 3e) indicate a weakening of the ion-dipole interaction, an observation strongly corroborated by our MD simulations (Fig. 3h), which calculate a significant decrease in the PBA-ion interaction energy. Likewise, the TDRS spectra (Fig. 3f) suggest an enhancement of the ion-ion Coulomb forces, a phenomenon perfectly mirrored in our simulations where the [Emim]⁺-[TFSI]⁻ interaction becomes dominant. Taken together, these tightly coupled experimental and theoretical results provide compelling evidence for an entropy-driven phase separation mechanism."
2. The manuscript demonstrates excellent stability over 100 heating-cooling cycles (Supplementary Fig. 20) and stability after 24 hours of continuous heating (Supplementary Fig. 21). While this is sufficient for a laboratory evaluation, real-world applications like smart windows require the material to endure much longer periods (several years) of UV irradiation and temperature fluctuations. The authors are advised to add a brief perspective or preliminary assessment in the discussion section regarding the material's long-term chemical stability under UV irradiation.
3. In the active optical display section (Fig. 6), the authors demonstrate both direct and indirect Joule heating. The main text or figure captions should explicitly state the voltage, current, or power density required to achieve the corresponding temperatures. For example, the SI mentions that a voltage of 180V and a frequency of 10 kHz are used to achieve global opacity.
4. To broaden the context and impact of the manuscript, the authors might consider expanding the Introduction to include

more foundational literature on ionogel science. While the review of smart windows is relevant, acknowledging the pioneering work on thermo-sensitive gels by Yongjun Men (Donghua University), high-strength ionogels by Feng Yan (Soochow University), and related supramolecular systems by Minghua Liu (Beihang University) would provide a more complete scientific background for the achievements presented here.

Version 1:

Reviewer comments:

Reviewer #1

(Remarks to the Author)

The authors have made extensive revisions by addressing all the comments and provide additional experimental results. Now the statements and conclusions are well supported by experimental data and proper analysis. It is recommended for publication in Nature Communications in its current form.

[Editor's Note: This Reviewer was asked to evaluate the response to Reviewer #2's comments, and provided the following:]

The authors have tried to respond to all the previous comments from Reviewer #2. However, some critical issues remain.

1, The SAXS and WAXS results indicate a phase separation domain size of about 7.2 nm, and some unspecified characteristic dimension of 4.5 Å and 7.2 Å. These dimensions could not result in any severe scattering of visible-NIR light that causes a transparency-to-opaque transition.

2, On the other hand, the authors' explanation to the dynamic phase separation is based on the ion-dipole interaction at low temperatures and its transition to ion-ion transition at high temperatures, which may violate some basic physics. Once again, the origin of LCST of the ionogels is not well established.

3, The correlation between formulation and performance is not well investigated. The authors only provide on the LCST values of different formulations. Some other important properties, including mechanical properties, low temperature transparency, high temperature transmittance, etc, should also be investigated, compared, and discussed. This will help readers better understand the materials design and proposed mechanisms.

The authors are requested to resolve these fundamental problems and provide more solid experimental data before it can be recommended for publication in Nature Communications.

Reviewer #3

(Remarks to the Author)

The authors have fully addressed all my concerns and have improved the quality and robustness of the manuscript through these revisions. I believe the work is highly innovative and meets the standards for publication. I recommend the acceptance of this manuscript in its current form.

Version 2:

Reviewer comments:

Reviewer #1

(Remarks to the Author)

The X-ray scattering/diffraction results show some ordered/crystalline structures, and the dimension of about 7 nm cannot cause opaque or translucent appearance of the hydrogels. The authors hypothesize that the 7 nm objects may aggregate to cause opaqueness. But no scattering data support this key hypothesis. I would like to recommend it to Nature Communications only when this important fundamental issue is supported by solid experimental data that verify the formation of phase separated structures.

Version 3:

Reviewer comments:

Reviewer #1

(Remarks to the Author)

The manuscript can be accepted for publication in Nature Communications in its current form.

Point-by-Point Response to Reviewers' Comments

Manuscript ID: NCOMMS-25-68161A

(Comments in black, responses in light blue, revised texts in *blue italics*)

We express our sincere appreciation to the editor and reviewers for their thorough review of our manuscript and for providing valuable and insightful feedback. In response to each comment, we have provided detailed point-to-point responses below, and we hope that the editors and reviewers will find them satisfactory. Thank you again for your time and expertise in reviewing our work.

Reviewer #1 (Remarks to the Author)

This manuscript reports on the fabrication of an asymmetric thermochromic ionogel with a bilayer structure comprised of a thermoresponsive light-scattering layer and a tough supporting layer. The thermochromic layer serves as the active layer to modulate light transmittance upon temperature changes. The main thermochromic mechanism lies in the thermoresponsive phase separation of IL and PBA at temperatures higher than its LCST. Meanwhile, the microphase-separated structures of P(BA-co-NIPAM) serve as energy dissipation mechanism to yield a tough ionogel. The bilayer can shift from transparent to opaque states as the temperature shuttles between temperatures below and above the LCST. The thermochromic property is utilized to modulate solar radiation through windows, which is demonstrated to lower the room temperature. The ion conductive ionogel shows Joule heating characteristic, which is used to motivate phase transition through electro-stimulation. This research is recognized as an interesting contribution to the field of smart materials with switchable and tunable optical properties. Although the demonstration to potential applications is fascinating, the fundamental sections on the materials design, synthesis, and characterizations are not clear or adequate to establish the new concept. Besides, the biomimetic design principle is not correct or necessary. The manuscript needs extensive revisions before it could be considered for publication in Nature Communications.

Response: We would like to express our deep gratitude to the reviewers for their expertise and the valuable time they dedicated to our manuscript. We sincerely

appreciate the constructive feedback, which has helped us significantly improve the manuscript.

In particular, we have paid close attention to the comment regarding the biomimetic design principle. Upon careful reconsideration, we agree with the reviewer that the original framing was not sufficiently reasonable or necessary for the core narrative of the work. Consequently, we have thoroughly removed all claims and discussions related to the biomimetic design principle throughout the manuscript. This revision helps sharpen the focus on the material's intrinsic properties and performance, and we believe the manuscript now presents a more accurate and convincing story. Meanwhile, we have also extensively revised the sections on materials design, synthesis, and characterization to provide clearer explanations and additional data where needed, ensuring the logic and novelty of this work are firmly established. All modifications have been highlighted in the revised manuscript for your convenience. Below, we provide a point-by-point response to each comment. We are grateful for the opportunity to revise the manuscript and hope that the changes we have made adequately address the reviewers' concerns.

1. The biomimetic concept is not correct or necessary. The asymmetric bilayer structure has nothing to do with the beetle carapace. No resemblance could be found between the current bilayer ionogel and the composite bilayer structure of the carapace. Such an analogue in fact harms the main idea of this study.

Response: We sincerely thank the reviewer for this critical comment regarding the biomimetic analogy in our manuscript. We have carefully reconsidered this perspective and agree that forcing a direct comparison with the beetle carapace was not essential to highlight the core innovation of our work, and may indeed have distracted from the primary focus on the material's intrinsic properties. In direct response to this suggestion, we have thoroughly revised the manuscript to remove all claims and discussions that directly linked the asymmetric bilayer structure of our ionogel to the specific composite Janus structure of the beetle carapace. The biomimetic narrative has been completely eliminated.

The focus of the manuscript has been reframed to concentrate squarely on the material's structural innovation, its outstanding performance (including mechanical toughness, environmental resilience, and dual-mode optical switching), and the underlying mechanism of the "thermosensitive dynamic phase separation" itself. We believe this shift in narrative strengthens the manuscript by placing undivided emphasis on the novel material design and its experimentally demonstrated capabilities, which stand on their own merits. The revised **Figure 1** is shown below:

Fig. 1: Design strategy and performance of ATIs. *a* Schematic of the cross-section of an environmentally adaptable ATI in which the top layer serves as a light scattering layer and the bottom layer serves as a tough supporting layer. *b* Two polymer networks in asymmetric ionogels. Left, dynamic phase-separated polymer network in ATI-B; right, in situ phase-separated copolymer network in ATI-BN. *c* Demonstration of the tough mechanical properties of ATI; inset shows glass (left) and ATI (right) after being crushed by a 1.25 t vehicle. *d* Optical transparency of ATIs in extreme cold (-70°C) and common high-temperature environments (ambient warming or Joule heating, 40°C), respectively; the inset shows transmittance of light at 550 nm wavelength. *e* A comprehensive performance comparison between ATIs and reported TCMs in multiple dimensions, including transparency, low-temperature resistance, self-supporting, stretchability, solar modulation capability, and adhesion. *f* ATI as a flexible optical switch mounted on glass: locally heated region (left) serves as a projection screen (low transmittance, a bird image serves as a projection object), while the ambient-temperature region (right) maintains high transparency for unobstructed visibility.

We are grateful to the reviewer for this insightful feedback, which has helped us to present our work in a more precise and compelling manner. We hope the revised manuscript now meets the reviewer's approval.

2. Some important terms, including phase separation and dynamic phase separation, are used without a clearly and precise definition. It is difficult to understand them in the ionogels and the thermochromic behavior.

Response: We sincerely thank the reviewer for this insightful comment regarding the need for clearer definitions of the terms "phase separation" and "dynamic phase separation" in the context of our ionogel system. We agree that providing precise definitions is essential for readers to accurately understand the distinct roles of these processes in the material's behavior. In response to this suggestion, we have thoroughly revised the manuscript to explicitly define and differentiate the two types of phase separation involved:

Thermosensitive dynamic phase separation: This refers to the reversible, temperature-induced phase separation occurring in the light-scattering layer (ATI-B), which is driven by the lower critical solution temperature (LCST) behavior and is responsible for the thermochromic switching.

In situ phase separation: This refers to the irreversible, solubility-driven phase separation during the polymerization of the tough support layer (ATI-BN), which creates the microphase-separated structure essential for energy dissipation and mechanical robustness.

We have introduced these precise definitions in the Introduction and Results sections and have consistently applied the clarified terminology throughout the revised manuscript to avoid ambiguity. We believe this clarification significantly enhances the conceptual clarity and helps readers better understand the synergistic interplay between the two layers in the ATI. Examples of partial revisions are as follows:

“This asymmetric thermochromic ionogel (ATIs) with exceptional mechanical robustness integrates: i) a thermoresponsive light-scattering ionogel layer engineered via thermosensitive dynamic phase separation (a temperature-mediated reversible process), and ii) a high-strength ionogel supporting layer fabricated by in situ phase separation (irreversible process)—the two covalently bonded at the interface through sequential photopolymerization (Fig. 1a).”

We are grateful for this suggestion, which has undoubtedly improved the precision and readability of our manuscript.

3. LCST is a most important characteristic of the ATIs (particularly for ATI-B). But this LCST is not clearly defined or explained. What is the mechanism? How to determine LCST? The DSC curve shows two transitions and the author assign the one at 34C as the LCST. What is the criterion? Supplementary Figure 17 shows different transition temperatures of ATI-Bs with different formulations. But why should the author identify them as LCST? How to tune the LCST (if it is true)? This section is very important for the whole study.

Response: Thank you for your thoughtful comments on our manuscript. We appreciate your questions regarding the LCST behavior of our ATI-B material, which is indeed a critical aspect of our study. Below, we provide a point-by-point response to address your concerns:

i): Mechanism of LCST

The LCST behavior is a hallmark of thermoresponsive polymers, where the polymer remains soluble in a solvent at low temperatures but undergoes phase separation upon heating, leading to the formation of a diluted polymer phase and a concentrated polymer phase. This phenomenon is primarily driven by entropy changes and the balance of interactions between the polymer and solvent. In the case of ILs as solvents, such as in our ATI system, the mechanism involves a delicate equilibrium of ion-dipole interaction, hydrogen bonding, electrostatic forces, and hydrophobic effects (*Polymer Chemistry*, 2024, 15, 2719-2739). The LCST behavior in ionogels—networks of polymers swollen with ILs—is primarily an entropy-driven process governed by the delicate balance between polymer-polymer, polymer-solvent, and solvent-solvent interactions. Below the LCST, the system is homogeneous due to favorable polymer-IL interactions, such as hydrogen bonding and ion-dipole forces. As temperature increases, the entropy term ($T\Delta S$) dominates the free energy change ($\Delta G = \Delta H - T\Delta S$), leading to a decrease in the overall solubility of the polymer in the IL and triggering phase separation (see **Supplementary Note 5 for details**).

“Supplementary Note 5. Entropy-driven dynamic phase separation in ATI-B.

Numerous studies report that hydrogen bonding and cation- π interactions can induce ordered structures between ILs and polymers, thereby generating a negative mixing entropy (ΔS_{mix}). Similarly, we propose that oriented solvation driven by ion-dipole interactions in our system could serve as the driving force for the observed lower critical solution temperature (LCST) behavior. At room temperature, the Gibbs free energy of mixing ($\Delta G_{mix} = \Delta H_{mix} - T\Delta S_{mix}$) is negative. However, heating endothermically disrupts these ion-dipole interactions, making ΔH_{mix} less negative. Concurrently, the

system gains entropy, increasing the magnitude of the entropic contribution ($T|\Delta S_{mix}|$). When $T|\Delta S_{mix}|$ exceeds $|\Delta H_{mix}|$, ΔG_{mix} becomes positive. This results in entropy-driven phase separation, where the ILs segregate into discrete domains. ”

ii): Identification and Determination of LCST

In response, we have provided a detailed explanation in **Supplementary Note 2** of the manuscript, outlining the methods used to determine and verify the LCST of the ATI-B layer. Our approach combines thermal and optical analyses to ensure robust identification of the phase transition temperature. The LCST was primarily identified using Differential Scanning Calorimetry (DSC). The DSC curve for ATI-B exhibited a distinct endothermic peak near 34°C, which is absent in the non-thermoreponsive ATI-BN control sample. This peak corresponds to the disruption of ion-dipole interactions between the polymer network and the ionic liquid solvent, leading to polymer phase separation. The process is endothermic due to the energy required to break these interactions, resulting in a measurable enthalpy change. Notably, the relatively small magnitude of the peak reflects the dynamic nature of the phase separation in ATI-B, which involves a subtle balance of interactions rather than a large-scale solvent expulsion event (the endothermic phenomenon in the DSC of ATI-B is similar to that reported in the literature: *Advanced Functional Materials*, 2023, 33, 2307240). To verify the DSC findings, we performed visible-wavelength transmittance tests (**Supplementary Fig. 16**). A sharp decrease in light transmittance was observed as the temperature surpassed the LCST, indicating a transition from a transparent to an opaque state due to light scattering from the phase-separated domains. This optical response is a hallmark of LCST behavior and aligns perfectly with the thermal transition detected by DSC. The combination of these techniques—DSC providing the thermodynamic signature and transmittance measurements capturing the optical manifestation of phase separation—offers compelling and consistent evidence for the LCST in ATI-B. We have revised the manuscript to clarify this multi-method approach and believe it firmly establishes the LCST characterization.

“Supplementary Note 2. LCST behavior of ATI-B.

The ATI proposed in this paper, particularly ATI-B, is a polymeric material with smart properties. The defining characteristic of thermoresponsive polymers lies in their response to temperature changes. Crucially, these polymers exhibit significant changes in solubility at their phase transition temperatures, primarily due to the precise compatibility window within their binary polymer/solvent phase diagrams. Based on their thermoresponsive behavior, these polymers are primarily categorized into two types: low critical solution temperature (LCST) behavior and high critical solution

temperature (UCST) behavior. LCST behavior implies that the polymer remains soluble in the solvent before reaching the phase transition temperature. Beyond this temperature, two distinct phases emerge—a diluted polymer phase and a concentrated polymer phase. Essentially, LCST characterizes the lowest point of phase diagram for a miscible system, beyond which only a single phase can exist, irrespective of polymer concentration (**Supplementary Fig. 20a**)¹³. Differential scanning calorimetry (DSC) is a commonly used method for determining the LCST of thermosensitive polymers, as their phase transitions are often accompanied by endothermic events¹⁴. DSC measurements were conducted on ATI-B and ATI-BN. The results indicated that, besides a step-like shift in the baseline, ATI-B displayed an additional endothermic peak near 34 °C, which was absent in ATI-BN. The step-like shift corresponds to the glass transition temperature (T_g) of the polymer. Within this temperature region, the specific heat capacity of the polymer increases, leading to greater heat absorption and a shift of the baseline toward the endothermic direction. The weak endothermic peak is ascribed to the LCST behavior of the polymer (**Supplementary Fig. 20b and 20c**). At this temperature, the interactions between the polymer and the IL solvent are disrupted, leading to precipitation of the polymer from the solvent. This process is endothermic, producing a small enthalpy change. This is because in ATI-B, the dynamic variation in ion-dipole interactions between the polymer and ionic liquid is insufficiently large to be classified as a large-scale phase separation phenomenon accompanied by significant solvent expulsion (with large enthalpy change). Consequently, the endothermic peak on the DSC curve is not prominent, which is consistent with previous reports¹⁵. Moreover, a key characteristic of LCST behavior is a significant alteration in the material's light-scattering capability, aligning with our observations in visible-wavelength transmittance tests (**Supplementary Fig. 16**).

Supplementary Figure 16. The transmittance of ATI within the visible light range at temperatures between 20-45°C.

Supplementary Figure 20. LCST behavior in the temperature range of -100 to 100°C. a) LCST behavior of ATI-B. b) DSC curve of ATI-B and c) ATI-BN. ”

iii): Tuning of LCST

The LCST of ionic gels is primarily regulated by adjusting the ratio of polymer to ionic liquid solvent. As shown in **Supplementary Fig. 21 and Table 2**, a lower polymer content tends to result in a lower LCST temperature for the ionic gel. This finding is consistent with the outcomes of numerous influential studies (*Soft Matter*, 2012, 8, 8067-8074; *Materials Horizons*, 2024,11, 3825-3834, *Journal of Materials Chemistry A*, 2023,11, 9626-9634).

Supplementary Figure 21. DSC curves of ATI-B in different polymer/IL ratios.

Supplementary Table 2. Formulations of ATI-B with different LCSTs.

Group	Additions (Monomers in mmol)				LCST (°C)
	BA	[Emim][TFSI]	1173	EGDMA	
I	19.5	6.4	0.05 mol%	0.05 mol%	28
II	23.4	5.1			34
III	27.3	3.8			38
IV	31.2	2.5			41

4. Concerning the SAXS and WAXS results, the authors attribute the scattering peaks to the contraction of the ionic clusters and the increase of pi-pi stacking distance caused by enhanced ion-ion interactions. This is quite arbitrary. It is not clear why should the ion-ion interaction be enhanced. What are the so-called ionic clusters? The authors claim the formation of bicontinuous structures in the ATI-BNs. But the SAXS pattern does not provide any useful information on the phase separated structures of ATI-BNs.

Response: We sincerely thank the reviewer for these insightful comments regarding the interpretation of our SAXS/WAXS data and the mechanistic explanation of ion-ion interactions. They have prompted us to provide a more rigorous and detailed explanation, supported by additional analysis. Our point-by-point responses are below.

i) Response to the comment on the enhancement mechanism of ion-ion interactions:

We agree that a clearer mechanistic explanation is crucial. Our revised discussion now explicitly outlines the entropy-driven process that leads to the strengthening of Coulombic interactions between ions upon heating above the LCST. In the LCST-type ionogel system, the IL cations and anions are initially stabilized within the polymer network primarily through relatively weak, non-directional ion-dipole interactions with the polymer chains (*CCS Chemistry, 2025, 0, 1-22*). This configuration contributes to the material's excellent low-temperature tolerance by suppressing IL crystallization (*Nano Energy, 2019, 63, 103847*). However, when the temperature exceeds the LCST, the polymer chains undergo a phase transition and collapse. This process is entropy-

driven, leading to the dissociation of these ion-dipole interactions. The freed ions, specifically the [Emim]⁺ cations and [TFSI]⁻ anions, then reassociate. Crucially, this reassociation is driven by a net increase in system entropy, favoring the formation of stronger and more concentrated ion-ion (Coulombic) pairs or clusters. This mechanistic explanation is strongly corroborated by multiple, tightly coupled experimental and theoretical results presented in the revised manuscript:

The temperature-dependent FTIR spectra show a redshift in characteristic peaks, indicating a weakening of the ion-dipole interaction between the IL and the polymer. Temperature-dependent Raman spectroscopy (TDRS) reveals a shift of the $\nu(-N-S)$ peak to lower wavenumbers, directly signaling enhanced Coulomb interaction between [Emim]⁺ and [TFSI]⁻. Molecular dynamics (MD) simulations quantitatively confirm that the [Emim]⁺-[TFSI]⁻ interaction energy increases with temperature and becomes the dominant interaction above the LCST. Two-dimensional correlation infrared (2DCOS FTIR) analysis further suggests that the phase separation is triggered by the preferential dissociation of anions from the polymer, followed by cation re-association. Therefore, the enhancement of ion-ion interaction is not arbitrary but is a direct consequence of the entropy-driven breakdown of the initial ion-dipole network and the subsequent reorganization of ions into more stable ionic clusters. We have revised the text to present this logical sequence more clearly.

“The microscopic interactions in thermosensitive dynamic phase separation were analyzed by in situ characterization and computational chemistry techniques. The temperature-dependent FTIR results (Fig. 3e) indicate a redshift of the characteristic peaks of ATI-B (from 1052 to 1050 cm⁻¹ and 1348 to 1346 cm⁻¹), suggesting a weakening of the ion-dipole interaction between the IL and polymer⁵¹. This observation is strongly corroborated by density functional theory (DFT), which calculate a significant decrease in the PBA-ion interaction energy per unit area (Supplementary Fig. 30). Likewise, the temperature-dependent Raman spectroscopy (TDRS) reveals a shift of the $\nu(-N-S)$ peak (Fig. 3f, Supplementary Fig. 31) to lower wavenumbers, implying enhanced Coulomb interaction between [Emim]⁺ and [TFSI]⁻⁵². This phenomenon is perfectly reflected in the molecular dynamics (MD) simulation, where the [Emim]⁺-[TFSI]⁻ interaction energy increases with rising temperature and becomes dominant (Fig. 3g). Furthermore, two-dimensional correlation infrared spectroscopy (2DCOS FTIR) analysis suggests that the phase separation is triggered by the preferential dissociation of anions from the polymer, followed by cation re-association (Supplementary Fig. 32, Table 3 and Note 4). Taken together, these tightly coupled experimental and theoretical results provide compelling evidence for an entropy-driven dynamic phase separation mechanism above the LCST, wherein enhanced Coulomb

forces promote the aggregation of solvent molecules detached from the polymer (Supplementary Note 5)⁵³. ”

We have also revised the discussion on WAXS in the original text as follows:

“The variable-temperature WAXS results of ATI-B reveal that the characteristic dimensions of its scattering peaks shift from 4.5 Å and 7.7 Å to 4.6 Å and 7.2 Å (Fig. 2d-ii, Supplementary Fig. 14 and Table 1). This change can be attributed to the increase in polymer chain spacing due to thermal expansion effects, coupled with the entropy-driven contraction of [Emim]⁺–[TFSI][–] ion clusters⁴¹. ”

ii) Response to the comment on the SAXS analysis of the bicontinuous structure in ATI-BN:

We sincerely thank the reviewer for raising this critical point regarding the evidence for the bicontinuous structure in the ATI-BN layer. We agree that a more rigorous analysis of the SAXS data is necessary to substantiate this claim, and we have undertaken substantial additional work to address this concern comprehensively. In direct response to the comment, we performed a new series of SAXS measurements specifically on the standalone ATI-B and ATI-BN layers. To move beyond qualitative observation and extract quantitative structural parameters, we conducted detailed fitting of the SAXS curves using the Debye-Anderson-Brumberger (DAB) model within the SASfit software (version 0.94.12). The DAB model is particularly well-suited for analyzing the characteristic size scales in two-phase systems with correlation length distributions, as it assumes smooth interfaces between phases—a key feature consistent with the proposed bicontinuous structure in the ATI-BN support layer. Through this quantitative fitting, we determined a phase separation feature size of approximately 7.3 nm for the ATI-BN layer. This specific result is now reported in the revised manuscript (Supplementary Fig. 13) and provides a concrete, numerically-grounded parameter describing its nanostructure.

To complement this quantitative SAXS analysis and provide multi-faceted evidence, we have also supplemented the manuscript with additional characterization data. We included clearer scanning electron microscopy (SEM) images that distinctly reveal the morphological difference at the interface between the ATI-B and ATI-BN layers, offering direct visual evidence of their distinct structures. Furthermore, we have added a schematic diagram (Supplementary Fig. 13) proposing the envisioned bicontinuous structure based on the consolidated evidence from SAXS and SEM. For contrast, the SAXS results for the active layer, ATI-B, confirm its high homogeneity on the 1-50 nm scale, with no scattering rings observed in the 2D pattern. This effectively rules out

internal phase separation within ATI-B and highlights the unique structural role of the ATI-BN layer. We have revised the relevant content in the 'Results' section to present these comparative findings and their interpretation more clearly and cautiously.

We believe that the combination of 1) quantitative model-based analysis of SAXS data (DAB model fitting), 2) direct morphological evidence (SEM), and 3) a proposed structural model (schematic diagram) now provides a robust and convincing support for the existence of a nanoscale bicontinuous structure in the ATI-BN layer. We are grateful for the reviewer's insightful suggestion, which has significantly strengthened the credibility of our analysis and discussion in this part.

“Results

The SAXS results show that ATI-B is highly homogeneous on the 1-50 nm scale, and the absence of scattering rings in the 2D SAXS spectrum proves that there is no any internal phase separation (Fig. 2d-i). In contrast, ATI-BN exhibits more pronounced scattering rings, and its SAXS spectrum displays distinct step-like peaks. Through computational fitting, its phase separation size was determined to be approximately 7.3 nm (Supplementary Fig. 13)⁴⁰.

Methods

*Perform fitting calculations on SAXS results using SASfit software (version 0.94.12) and Origin software (version 9.9.0.225). The phase separation size in ATI-BN was characterized using the Debye-Anderson-Brumberger (DAB) model. The appropriateness of this model for the determination of correlation lengths in random, non-granular two-phase systems is confirmed by its assumption of smooth interfaces, a feature consistent with the bicontinuous structure of ATI-BN. The pair correlation function $\gamma(r)$ is given by **Equation 1**:*

$$\gamma(r) = \exp(-r/\xi) \quad (1)$$

where r is radial distance.

*The macroscopic scattering cross-section in the DBA model is given by **Equation 2**:*

$$I(q) = \frac{(\pi\Delta\eta\xi^3)^2}{(1 + q^2\xi^2)^2} \quad (2)$$

where q is scattering vector; $I(q)$ is scattering intensity; ξ is correlation length; $\Delta\eta$ is scattering length density contrast.

*Bragg spacing in WAXS is given by **Equation 3**:*

$$d = \frac{2\pi}{q} \quad (3)$$

where d is the real-space feature distance; q is the scattering vector, defined by the **Equation 4**:

$$q = \frac{4\pi\sin\theta}{\lambda} \quad (4)$$

where θ is half-scattering angle; λ is wavelength of the incident X-rays.

Fig. 2: Preparation, composition, and characterization of asymmetric structures of ATIs. *a* ESP simulations of key ATI components: polymer monomers BA and NIPAM, and IL solvent [Emim][TFSI]. *b* FTIR spectra of solvent-free PBA, pure IL, and ATI-B. *c* Microscopic image (left) and SEM image (right) of the cross-section of the ATI. *d* 2D spectra of SAXS (i) and WAXS (ii). *e* Raman spectra of ATI-B and ATI-BN; insets i, ii, iii are Raman mapping images of ATI-B, ATI-BN, and ATI cross sections, respectively. *f* Optical images of physical bilayer ionogel and ATI (upper PBA ionogel layer, stained with metanil yellow; lower P(BA-co-NIPAM) ionogel layer, stained with brilliant green) and schematic of the proposed interface. *g* Characterization of the rheological properties of ATIs, including amplitude sweeping (left), frequency sweeping (middle), and temperature ramping (right).

Supplementary Figure 13. Analysis of bicontinuous structure in ATI-BN. a) 1D SAXS spectra. b) SEM image of the interface of ATI, with ATI-BN (rough) on the left and ATI-B (smooth) on the right. c) The proposed schematic diagram of the bicontinuous structure and phase-separated correlation length in ATI-BN. ”

5. The interface robustness is key to the bilayer structure. The interface strength should be tested.

Response: We sincerely thank the reviewer for this critical comment regarding the interface robustness of ATI bilayer structure. We fully agree that quantifying the interfacial strength is essential for evaluating the mechanical integrity and practical applicability of the material.

In our original manuscript, we had included data on interface strength in **Figures 4e** and **4f**, but we acknowledge that the rationale and methodology may not have been sufficiently emphasized. Due to the significant mechanical property disparity between the thermally responsive layer (soft) and the support layer (tough) in the ATI, conventional methods like T-peel testing are unsuitable, as they would induce cohesive failure within the softer layer rather than accurately measuring the interfacial adhesion strength. To address this, we employed the lap shear test—a widely recognized and robust method for quantifying interfacial shear strength in systems with mismatched mechanical properties (**Figure R1** and **Figure 4e**). This approach is explicitly recommended for evaluating bonded assemblies where peel tests are impractical. The quantitative results from this test, presented as the shear stress-displacement curve in **Supplementary Fig. 37**, provide direct evidence of the interfacial shear strength. Furthermore, we conducted finite element simulations to model the interface peeling process, comparing the ATI with a physical bilayer ionogel (**Figure 4f**). The simulations indicate that stresses are higher in the ATI, with significant stress concentrations occurring at the interface joints, highlighting its superior interface toughness. These additions (**Supplementary Fig. 37** and **associated analysis**) now explicitly demonstrate that the covalent/entanglement interface in the ATI effectively resists delamination, aligning with the reviewer’s emphasis on interface robustness. We have

revised the main text to clarify the methodology and results, ensuring that the interface strength is presented rigorously.

Figure R1. Schematic diagram of the apparatus for lap shear test.

“he interfacial strength between ATI-B and ATI-BN was tested by lap-shear experiments, and the interfacial shear strength of ATI reaches 600 kPa, which is 110 times higher than that of the physical bilayer ionogel; and the strong interfacial strength of ATI shows a significant advantage even when compared with commercial adhesive tapes (Fig. 4e, Supplementary Fig. 37). We simulate the interfacial peeling process in physical bilayer ionogel and ATI using finite element analysis (FEA). The results show that under pulling of vertically upward force, ATI exhibits higher stress concentration than the bilayer ionogel, and the near-perpendicular angle between upper and lower layer indicates that the interfacial bonding of ATI is significantly stronger (Fig. 4f).

Fig. 4: Tough mechanical properties of ATI. Comparison of the mechanical properties of ATI and pure PBA ionogel. Including **a** Stress-strain curve, **b** toughness and Young's modulus; **c** optical photograph. **d** Comparison of load capacity in real-world scenarios, ATI can easily lift 1 kg of weight, which is not possible for pure PBA ionogel. **e** Comparison of the interfacial shear stress of ATI with other adhesive materials, including commercially available 3M VHB, 3M 9080, and bilayer ionogel; inset shows a schematic of the lap-shear experimental setup for measuring interfacial strength. **f** FEA mechanical simulation of bilayer ionogel and ATI, where the darker color (red) is subjected to higher stress. **g** Puncture resistance performance of ATI. The appearance of ATI before (inset), during (left), and after (right) puncture is not significantly damaged. **h** Stress-strain curves of ATI under different compression cycles; insets show optical photographs before, during, and after compression, as well as stress-strain curves under 100 compression cycles. **i** Comparison of the properties of ATI with reported thermochromic materials, including tensile strength and effective LCST range. **j** Photographs of ATI being used on a shaped surface at 25°C (left) and 40°C (right).

Supplementary Figure 37. Shear stress-displacement curve, including ATI, 3M VHB, 3M 9080, and bilayer ionogel. ”

We are grateful for this insightful suggestion, which has strengthened our discussion on mechanical properties of ATI and enhanced the overall quality of the manuscript.

6. Most schematic illustrations look fancy. But some of them are over rendering. In figure 3a, for example, what is the ion-dipole interaction? What is the dipole? in Figure 3b, it is difficult to image that the light can be "scattered" throughout the opaque ionogel. It is violating the basic optical physics.

Response: We sincerely thank the reviewer for these insightful comments regarding the clarity and scientific accuracy of our schematic illustrations. We agree that ensuring these figures are both visually clear and physically precise is crucial for effectively conveying the core mechanisms of our work. We have carefully revised these two figures in response to these specific points.

i) Response regarding the ion-dipole interaction in Figure 3a:

We thank the reviewer for seeking clarification on the ion-dipole interaction depicted in this schematic. The dipole in question originates from the permanent dipole moment of the carbonyl group (C=O) within the poly (n-butyl acrylate) (PBA) polymer network. The oxygen atom in the carbonyl group is highly electronegative, creating a partial negative charge (δ^-), while the carbon atom carries a partial positive charge (δ^+). This establishes a permanent electric dipole moment within the functional group of the polymer chain.

The interaction occurs when a $[\text{Emim}]^+$ from the IL is electrostatically attracted to the partially negative oxygen atom of the carbonyl group. This ion-dipole force is a key interaction that contributes to the stability of the ionogel by helping to integrate the IL

within the polymer network below the LCST. In the revised version of Figure 3a, we have now explicitly drawn the relevant chemical groups (specifically highlighting the carbonyl groups in the PBA chain) and illustrated the local coordination environment between the IL cations and these dipoles. This revision aims to make the origin of the dipole and the nature of the interaction visually unambiguous and directly tied to the chemical structure of our material.

ii) Response regarding the depiction of light scattering in Figure 3b:

We appreciate the reviewer's valid criticism regarding the depiction of light scattering in the original schematic diagram. We acknowledge that the previous illustration could be misinterpreted in a way that seemed to violate basic optical principles concerning opaque materials.

The intended phenomenon we wished to illustrate is indeed the multiple scattering that occurs due to refractive index mismatch. When the temperature rises above the LCST, the homogeneous ATI-B layer undergoes phase separation, leading to the formation of distinct, nanoscale domains: polymer-rich regions and IL-rich domains ("islands"). The key point is that these domains have different refractive indices. When the incident light propagates in this heterogeneous microstructure, it is not simply absorbed; on the contrary, it undergoes extensive multiple scattering (refraction and reflection) at the numerous interfaces between these domains. During this process, the direction of light propagation is randomized, making the material appear opaque to the observer. In the revised Figure 3b (**new Figure 3c**), we have completely redesigned the schematic diagram to accurately depict this mechanism. The new diagram shows: i) A heterogeneous internal structure with distinct domains after phase separation. ii) The incident light entering the material. iii) The propagation path of the incident light being randomized through multiple scattering at the domain interfaces, ultimately preventing it from passing through in a straight line.

We believe this revised figure now correctly and effectively conveys the physical origin of the switching between transparent and opaque states, which is rooted in light scattering from a heterogeneous microstructure, and does not violate the principles of optical physics. We are grateful for the reviewer's keen insight, which has significantly improved the accuracy of our schematics.

“The thermochromic behavior of ionogels originates from dynamic phase separation within ATI-B. At low temperatures, the IL remains uniformly dispersed in the PBA matrix via ion-dipole interactions (primarily cations and the -C=O group in ester moieties), maintaining a homogeneous single phase (Fig. 3a)⁴⁶. With increasing

temperature above the LCST, uniformly distributed anions and cations aggregate around the polymer network⁴⁷, forming distinct liquid domains that result in a two-phase system (Fig. 3b). When dynamic phase separation occurs, a large number of “island-like” IL domains inside ATI-B scatter light, resulting in an opaque state (Fig. 3c).

Fig. 3: Dynamic phase separation principle and microscopic interaction analysis of ATIs. *a* Proposed schematic of the thermosensitive dynamic phase separation in ATI-B. *b* Optical microscopy images of ATI during the heating process. *c* Proposed schematic of the propagation path of light inside ATI-B below (left) and above (right) the LCST. *d* Optical photos of ATI under different temperatures. *e* Temperature-dependent FTIR spectra. *f* Temperature-dependent Raman spectra (colour scale: red, high intensity; blue, low intensity). *g* MD snapshots of PBA and [Emim][TFSI] at 298K and 313K. The calculated interaction energies of ion-dipole interactions (PBA-[Emim]⁺, PBA-[TFSI]⁻) and ion-ion interactions ([Emim]⁺-[TFSI]⁻) in ATI at 298K and 313K; insets show the ESP of the optimized structure of [Emim]⁺-[TFSI]⁻ at the two temperatures (colour scale: red, negative; blue, positive).”

7. It is very important to separately characterize the optical properties and transitions of ATI-Bs and ATI-BNs. So far, no picture or systematic data are presented for these single layers. In most cases, photos of ATIs are presented, so it is difficult to identify the contribution to the optical transition.

Response: We sincerely thank the reviewer for this insightful comment regarding the necessity to separately characterize the optical properties of the individual layers. We agree that distinguishing the contribution of each layer is crucial for a deeper understanding of the optical transition mechanism in ATI.

In direct response to this suggestion, we have conducted and now present a systematic comparative study of the standalone thermally induced light-scattering layer (ATI-B) and the mechanically tough support layer (ATI-BN). The new data, included as **Supplementary Fig. 18 and 19** and discussed in the main text, clearly reveal their distinct roles: The ATI-B layer exhibits excellent thermosensitivity, undergoing a significant and visually observable transition from transparent to opaque upon heating. This confirms its role as the active light-scattering component responsible for the optical switching. In contrast, the ATI-BN layer maintains high transparency and mechanical stability across the entire tested temperature range. This demonstrates its role as a passive, optically stable, and mechanically robust support, which ensures the structural integrity of the ATI without interfering with the thermochromic effect of the active layer.

We believe these separate characterization results have effectively clarified the respective contributions of the ATI-B and ATI-BN layers in ATI: the optical transition is dominantly driven by the phase separation within the ATI-B layer, while the ATI-BN layer provides essential mechanical support. We are grateful for the reviewer's suggestion, which has significantly strengthened the persuasiveness of the relevant discussions in our manuscript.

*“The transmittance of ATI-B in the visible light range decreases from more than 85% to less than 10% when the temperature increases from 20°C to 45°C, and haze within the visible light spectrum also increases significantly accordingly (**Supplementary Fig. 16 and 17**). In contrast, ATI-BN exhibits thermal stability within the same temperature range (**Supplementary Fig. 18 and 19**).*

Supplementary Figure 18. The changes in optical performance observed by heating ATI-B and ATI-BN separately within the temperature range of 25–50°C. Scale bar: 1 cm.

Supplementary Figure 19. Comparison of optical performance and mechanical appearance between ATI-B and ATI-BN under switching between 25°C and 50°C. Scale bar: 1 cm.

8. The solar modulation capability is low even at the visible range. This reminds of Figure 1g, where the authors compare the performances of the current TCMs with those reported in the literature. Unfortunately, the thermochromic and solar modulation performance is not included in this radar chart for some reason. It is tricky and unfair to mislead the readers.

Response: We sincerely thank the reviewer for this critical comment regarding the representation of solar modulation capability of our material and its comparison with existing technologies. We agree that providing a comprehensive and fair comparison is of utmost importance.

In direct response to the reviewer's valuable point, we have thoroughly revised Figure 1. We have now included "Solar modulation capability (ΔT_{sol})" as a key metric in the radar chart (**new Figure 1e**). This revision allows for a direct and transparent visual comparison between our ATI and other reported thermochromic polymers (TCPs) across all performance indicators, eliminating any possibility for ambiguity.

Furthermore, to provide a more quantitative and focused perspective on this specific property, we have added a new **Supplementary Fig. 40**. In this figure, we directly compares the solar transmittance change (ΔT_{sol}) and luminous transmittance change (ΔT_{lum}) of ATI with those of other state-of-the-art solar modulation materials reported in the literature.

The consolidated data from these revised and new figures confirm that while the solar modulation capability of our ATI may not represent a record-breaking value, it demonstrably remains competitive and falls within the advanced level among its peer materials. More importantly, this performance is achieved alongside other critical advantages highlighted in the radar chart, such as exceptional mechanical toughness, low-temperature tolerance, and self-adhesion, which are essential for practical applications but are rarely combined in a single material.

“A comprehensive performance comparison between ATIs and reported TCMs was conducted, evaluating transparency, low temperature resistance, self-supporting (without additional encapsulation), stretchability, solar modulation capability, and self-adhesion (Fig. 1e)^{21, 35, 36, 37, 38}. Notably, ATIs demonstrate superior low-temperature stability and mechanical strength while maintaining their solar modulation capability.

Fig. 1: Design principle and performance of ATIs. *a* Schematic of cross-section of an environmentally adaptable ATI in which the top layer is a light scattering layer and the bottom layer is a tough supporting layer. *b* Two polymer networks in asymmetric ionogels. Left, dynamic phase-separated polymer network in ATI-B; right, in situ phase-separated copolymer network in ATI-BN. *c* Demonstration of the tough mechanical properties of ATI; inset shows glass (left) and ATI (right) after being crushed by a 1.25 t vehicle. *d* Optical transparency of ATIs in extreme cold (-70°C) and common high-temperature environments (ambient warming or Joule heating, 40°C), respectively; the inset shows transmittance of light at 550 nm wavelength. *e* A comprehensive performance comparison between ATIs and reported TCMs in multiple dimensions, including transparency, low-temperature resistance, self-supporting, stretchability, solar modulation capability, and adhesion. *f* ATI as a flexible optical switch mounted on glass: locally heated region (left) serves as a projection screen (low transmittance, a bird image serves as a projection object), while the ambient-temperature region (right) maintains high transparency for unobstructed visibility.

Compared with the recently reported adjustable phase-change materials for energy-saving smart windows, ATI still demonstrates competitive solar modulation capabilities (Supplementary Fig. 40).

Supplementary Figure 40. Comparison of ATI's optical modulation capabilities (including ΔT_{lum} and ΔT_{sol}) with previously reported materials ^{S6, S7, S8, S9, S10}.

We are grateful for this suggestion, as it has guided us to present a more balanced, transparent, and comprehensive performance comparison, thereby strengthening the overall credibility of our work. We hope the reviewer finds these revisions satisfactory.

9. Figures 5h and 5i do not add much to the manuscript.

Response: We thank the reviewer for the valuable suggestion regarding the focus of the main text. Upon reflection, we agree that the specific aesthetic demonstration in the original Figures 5h and 5i could be streamlined or weakened to highlight the core idea of the paper. Therefore, we have moved these figures to the Supplementary Information (now Supplementary Fig. 42) and condensed the related discussion in the main text. Instead, we have introduced new building energy simulation data that directly quantifies the practical energy-saving effect of our smart windows, which we believe significantly enhances the application-oriented value of Figure 5. We have retained the aesthetic customization content in the Supplementary Information because we believe that the ability to tailor the color and shape of the ATI windows, as demonstrated therein, is a relevant aspect for potential real-world applications where architectural integration and design flexibility are important. We hope this revision, which prioritizes the central performance metrics in the main text while maintaining diversity in material design and application, can be approved by the reviewer.

“To comprehensively illustrate the energy-saving capabilities of ATI smart windows, we conducted an extensive simulation with EnergyPlus to evaluate the annual energy

*consumption of a building model fitted with ATI smart windows (**Supplementary Fig. 41**). Compared to clear windows with pure glasses, ATI smart windows deliver more significant energy savings in China's warmer urban areas, highlighting their potential for application in solar-regulated devices (**Fig. 5h**). For the aesthetic and personalized design of windows, ATI can be easily customized into complex window patterns or colors through simple reprocessing such as templating, cutting, dyeing, and modular splicing, without affecting its intrinsic thermally-induced transparency transition (**Supplementary Fig. 42**).*

Fig. 5: Passive smart cooling windows realized by ATI. *a* Proposed light modulation schematic for ATI smart windows. *b* Hydrophobicity of ATI smart windows, including contact angle (i) and adhesion (ii) of ATI-BN compared with glass and PNIPAM hydrogel, as well as simulated raindrop sliding test (iii). *c* Demonstration of autonomous adhesion of ATIs on different common transparent substrates. *d* UV-Vis-NIR transmittance spectra of ATI-based windows at 20–45°C, together with the solar irradiance intensity (air mass 1.5 global) and photopic luminous efficiency. *e* The temperature variation of model rooms with different windows under the illumination of simulated sunlight (xenon lamp) over time; inset is an optical photograph of the test site. *f* On-site optical photos of the model rooms during the cooling performance testing process. *g* Variation of room temperature in model rooms with different windows within 24 h (Xi'an, 34°15'N, 108°40'E, May 20, 2025). *h* China energy-saving map of the building model with ATI smart windows.

Supplementary Figure 42. Aesthetic customization by ATI. a) Custom display of the Chinese-style “Double Happiness” pattern on ATI, dyed with Rhodamine B. b) Modular coloring of ATI was performed using various dyes, including Rhodamine B, Soap Yellow, Toluidine Blue, and Brilliant Green. c) Demonstration of the thermochromic properties of dyed ATI. ”

10. For the Joule heating, what is the voltage used to heat the ionogel above its transition temperature? "Equivalent opacity can be achieved below the LCST via in situ Joule heating." What does this mean? Is the temperature of the ionogel below its LCST when it remains opaque? This statement is quite confusing and misleading. What is the merit or significance to use such an opaque "screen" for projection?

Response: We sincerely thank the reviewer for these insightful questions regarding the operational parameters and functional significance of ATI. The comments have helped us clarify key points in the manuscript and better articulate the value of our work. Our point-by-point responses are detailed below.

i) Response to the question on the operating voltage for Joule heating:

We thank the reviewer’s questions regarding the driving voltage. In our study, the voltage applied to achieve in-situ Joule heating and induce the transparency transition in the ATI is 180 V, with a frequency of 10 kHz. We acknowledge that omitting these specific parameters in the original manuscript was an oversight. In response to this question, we have now explicitly provided these values in the revised caption of **Figure 6**. Furthermore, to provide a deeper understanding of the electro-thermal relationship, we have conducted a supplementary investigation into the relationship between the

applied voltage and the temperature rise of the ATI. These new data, which illustrate the voltage-dependent heating kinetics, have been added as **Supplementary Fig. 45**. We believe these supplementary contents ensure that the experiments are fully reproducible and transparent.

*“Within 15 s after applying voltage (180 V, 10 kHz), the local temperature of the ATI reaches up to 55°C and undergoes a complete transparent-to-opaque transition (**Fig. 6e-i, Supplementary Movie 6**). Further testing indicates that the temperatures resulting from Joule heating on the surface of ATI is positively correlated with the applied AC voltage (**Supplementary Fig. 45**).”*

Fig. 6: Flexible dynamic optical display realized by ATI. **a** Schematic of a flexible dynamic display device driven by localized indirect heating. **b** Optical photographs of a localized optical display device. **c** Demonstration of localized optical displays, including optical photographs and thermal infrared images in the planar state (i), and localized optical displays on curved substrates (ii). The DC voltage for the conductive Joule heating layer is 24 V. **d** Schematic and equivalent circuit diagram of the ATI-based global Joule heating system. **e** Demonstration of in situ Joule heating of ATI using an AC power supply (180 V, 10 kHz), including optical and thermal infrared images before and after power on in the planar state (i), as well as thermal infrared images of ATI in the stretched and bent states when power on (ii). **f** Schematic of a smart projection display system based on ATI optical display switching; and **g** Realistic demonstration of the ATI smart projection display system, including the global projection of the famous painting of “Dance” and the traditional Chinese calligraphy “Ascending to the Height”, as well as localized magnified images (inset). **h** Demonstration of a dynamic localized projection display. The upper half is an opaque display area due to localized heating for displaying the weather conditions of the day, and the lower half is a highly transparent due to not being heated.

Supplementary Figure 45. The correlation between the surface temperature of ATI and the applied AC voltage. a) A photo from the test site shows the monitoring of ATI surface temperature using a single-channel T-type thermocouple. b) Relationship between ATI surface temperature and applied AC voltage; insets show optical photographs of ATIs at different voltages.

ii) Response to the confusion regarding the statement "Equivalent opacity can be achieved below the LCST via in situ Joule heating":

We apologize for the lack of clarity in our original phrasing, which is indeed confusing and even could be misunderstood. The intended meaning was that even if the ambient temperature is below the material's LCST, we can still actively switch the ATI from transparent to opaque state by employing the built-in Joule heating capability. This demonstrates the dual-mode switching (passive thermal response and active electrical control) of our ATI. To eliminate any ambiguity, we have revised the contentious sentence in the main text to: "When the ambient temperature is below the LCST of ATI, transparent to opaque transition can also be achieved through in situ Joule heating." We trust this revised expression accurately conveys the intended meaning and clarifies that the Joule heating actively raises the local temperature of the ATI above its LCST to trigger the phase transition and transparency change.

"When the ambient temperature is below the LCST of ATI, transparent-to-opaque transition can also be achieved through in situ Joule heating. Projections of classical artworks onto the opaque ATI surface maintained high color saturation and display resolution (Fig. 6g)."

iii) Response to the question on the merit of using such an opaque "screen" for projection:

We appreciate the reviewer's concern regarding the practical significance of this projection functionality. The core merit lies in expanding the functionality and application potential of building windows, which are large, flat surfaces that are ubiquitous in human daily life but have not been fully utilized traditionally. While most prior research on smart windows has focused almost exclusively on their energy-saving aspects (e.g., solar heat gain modulation), our work explores another possibility: transforming the window into an interactive optical display platform that extend the concept of "functional windows" beyond mere climate control. By enabling the window surface to serve as a projection screen for information display or ambient lighting, we aim to enhance its role in smart homes and intelligent buildings, increasing their value in information interaction and space utilization. As we now state in the revised discussion: "The above attempts not only provide innovative ideas for next-generation optical display devices but also address the shortcomings of existing smart windows that only have a single function." This approach aligns with emerging trends of modern buildings that seek to integrate dynamic digital displays seamlessly into building surfaces.

“The above attempts not only provide innovative ideas for next-generation optical display devices, but also address the shortcomings of existing smart windows that only have a single function.”

Reviewer #2 (Remarks to the Author)

This manuscript reports an asymmetric thermochromic ionogel (ATI) inspired by Janus structures: a thermally induced light - scattering layer (tunable LCST) is integrally coupled to a mechanically tough support layer via sequential radical polymerizations and in-situ phase separation. The device claims high optical contrast, low-temperature tolerance, high toughness, self-adhesion without edge sealing, and both passive and active optical switching, with demonstrations spanning model-building windows and patterned heating for simple displays. The thermal scattering and high toughness are considered through the covalent/entanglement interface of "dynamic thermally induced phase separation (scattering layer) in-situ phase separation (toughening layer)", which is rarely seen in PNIPAm hydrogel or single-phase ionic gel systems. The concept is compelling. To meet Nature Communications standards, the manuscript would benefit from stronger evidence connecting thermal, optical, and mechanical responses; from durability and lifetime statistics; and from explicit safety and energy metrics for device operation.

Response: We are sincerely grateful to the reviewer for the very positive and constructive feedback on our manuscript. We deeply appreciate the reviewer's recognition of our work as "compelling" and the valuable suggestions to further strengthen it. We fully agree that addressing these points is crucial to fully meet the high standards of Nature Communications. In accordance with the comments, we have performed additional experiments, analyses, and revisions to provide stronger mechanistic evidence, durability statistics, and explicit safety/energy metrics. Our point-by-point responses and the corresponding revisions in the manuscript are detailed below.

1. Given the ionogel's advertised customizability, I encourage the authors to expand the property space they present so that readers can see how formulation choices map onto performance.

Response: We sincerely thank the reviewer for this valuable suggestion regarding the customization of the ionogel's properties. We agree that explicitly demonstrating how formulation choices map onto performance is crucial for highlighting the material's adaptability. In response, we have expanded the property space analysis in the revised manuscript, as detailed in **Supplementary Fig. 21 and Table 2**. Our systematic investigation reveals a clear correlation between the polymer content in the ATI-B formulation and its LCST. Specifically, lower polymer content consistently leads to a

lower LCST, enabling precise tuning of the thermoresponsive behavior. This straightforward approach—adjusting the ratio of polymer to IL—allows for the design of ATIs with tailored LCSTs to suit diverse environmental conditions. For instance:

In regions with cooler average temperatures (e.g., North China), ATIs with a lower LCST can be optimized to activate their opaque, energy-saving state at relatively lower ambient temperatures. Conversely, in warmer areas, the LCST can be elevated to ensure optimal solar modulation and efficiency during hotter periods. By mapping formulation parameters (e.g., polymer/IL ratio) directly to performance metrics (LCST), we demonstrate how users can strategically customize ATIs for specific geographic or operational needs. This scalability and simplicity in tuning enhance the flexibility of material in practical applications, supporting its potential for widespread adoption in smart windows and other applications. We have incorporated this discussion into the manuscript to clarify the customization strategy and its implications.

“By varying the ratio of the polymer to the IL, the LCST of ATI-B can be tailored between 28°C and 41°C to meet specific requirements (Supplementary Fig. 21 and Table 2).”

Supplementary Figure 21. DSC curves of ATI-B in different polymer/IL ratios. ”

Supplementary Table 2. Formulations of ATI-B with different LCSTs.

Group	Additions (Monomers in mmol)				LCST (°C)
	BA	[Emim][TFSI]	1173	EGDMA	
I	19.5	6.4	0.05 mol%	0.05 mol%	28
II	23.4	5.1			34
III	27.3	3.8			38
IV	31.2	2.5			41

We believe these additions effectively address the reviewer's concern by illustrating the expandable property space and providing readers with a clear roadmap for performance optimization through formulation design.

2. Because the switching mechanism is fundamentally scattering-based, haze is the pivotal figure of merit for privacy glazing and projection/screen-like use. Please report haze as a function of wavelength and temperature across the visible range and, ideally, include angle-resolved measurements.

Response: We sincerely thank the reviewer for this insightful comment regarding the pivotal role of haze as a figure of merit for our scattering-based switching mechanism. We fully agree that comprehensive optical characterization, including the wavelength, temperature, and angular dependence of haze, is crucial for evaluating the material's performance in privacy glazing and projection-screen applications.

In direct response to this suggestion, we have supplemented and now report the haze as a function of wavelength across the visible spectrum (e.g., 380-780 nm) at key temperatures (e.g., below and above the LCST) in the revised manuscript (**Supplementary Fig. 17**). This data, obtained by measuring the total transmittance and the specular transmittance separately (*Advanced Functional Materials*. 2023, 33, 2305998), quantitatively demonstrates the dramatic change in light-scattering efficiency upon phase transition, which is also the core reason why the ATI-B is suitable for projection/screen-like use.

“The transmittance of ATI-B in the visible light range decreases from more than 85% to less than 10% when the temperature increases from 20°C to 45°C, and haze within the visible light spectrum increases dramatically from below 10% to around 94% (Supplementary Fig. 16 and 17).

Supplementary Figure 17. Variation of haze with temperature within the visible light wavelength range. ”

Regarding the reviewer's valuable suggestion for angle-resolved measurements, we deeply appreciate its importance for a complete optical analysis. However, we must respectfully explain a current technical limitation. Our spectrometer (QEPro, Ocean Optics, USA), while excellent for measuring total transmittance and haze according to established standards (such as ASTM D1003), is not equipped with the specialized goniometric accessories required for reliable, quantitative angle-resolved scattering measurements. Attempting to obtain such data without the proper fixture would compromise the accuracy and reproducibility of the results, as the collection of scattered light intensity is highly sensitive to the precise geometric alignment between the sample, the light source, and the detector. We have therefore chosen to refrain from reporting potentially unreliable angle-dependent data at this stage. We are committed to addressing this aspect in our ongoing research and plan to upgrade our spectroscopic capabilities to include such advanced characterization. We hope the reviewer finds our current wavelength- and temperature-dependent haze data, combined with this forthright explanation, satisfactory for the present study. We are grateful for this suggestion, which has important guiding significance for our future research.

3. How does the polymer ionogel system maintain a transparent phase below the LCST at deep sub-zero temperatures. To what extent is the observed stability a general property of ionic liquids, and can the authors compare hydrophobic, low-water-activity ILs with more hygroscopic ones to establish whether

hydrophobicity and low volatility are necessary conditions for low-temperature durability and frost-free transparency?

Response: We sincerely thank the reviewer for raising these critical points regarding the low-temperature behavior of our polymer ionogel system. We appreciate the opportunity to clarify the mechanisms and generality of the observed stability based on our experimental findings, as detailed in **Supplementary Note 3** of the revised manuscript. Below, we provide a point-by-point response to address the specific queries.

i) Mechanism of maintaining transparency at deep sub-zero temperatures

The polymer ionogel system maintains a transparent phase below the LCST even at deep sub-zero temperatures (down to -70°C) primarily due to the inherent properties of the ILs used. As elucidated in **Supplementary Note 3**, ILs exhibit structural asymmetry arising from bulky, asymmetric ions (e.g., imidazolium cations combined with anions like $[\text{TFSI}]^{-}$ or $[\text{BF}_4]^{-}$). This asymmetry inhibits regular crystalline packing upon cooling, suppressing crystallization and leading to low melting points and a propensity for supercooling. Consequently, the ILs remain in a liquid-like state without freezing or forming opaque crystalline phases. Additionally, physical interactions (e.g., ion-dipole forces) between the ILs and the polymer chains further inhibit crystallization, enhancing the low-temperature tolerance of the ionogel. The combination of these factors ensures that the system retains transparency and stability even under extreme cold conditions.

ii) Generality of stability across ionic liquids

The observed low-temperature stability is a general property of ionic liquids, not limited to hydrophobic types. To assess this, we compared four ILs with the same cation ($[\text{Emim}]^{+}$) but differing anions: $[\text{Emim}][\text{TFSI}]$ (hydrophobic), $[\text{Emim}][\text{BF}_4]$ (moderately hydrophilic), $[\text{Emim}]\text{Cl}$ (hydrophilic), and $[\text{Emim}][\text{EtSO}_4]$ (highly hydrophilic). Despite variations in hydrophobicity and hygroscopicity, all four IL-based ionogels demonstrated excellent low-temperature tolerance, maintaining transparency and performance at -70°C without freezing or opacity (**Supplementary Note 3**). This indicates that the stability is rooted in the fundamental ionic structure of ILs—particularly their asymmetry and low crystallization tendency—rather than being exclusive to hydrophobic ILs.

iii) Comparison of hydrophobic vs. hygroscopic ILs

To determine whether hydrophobicity and low volatility are necessary conditions for low-temperature durability, we explicitly compared hydrophobic ILs (e.g.,

[Emim][TFSI]) with hygroscopic ones (e.g., [Emim]Cl and [Emim][EtSO₄]). Key findings from **Supplementary Note 3** include:

Hydrophobicity and volatility: All ILs exhibited negligible vapor pressure at elevated temperatures, confirming that low volatility is a common trait. However, hydrophobicity varied significantly, with [Emim][TFSI] being immiscible with water (most hydrophobic) while others were miscible.

Low-temperature performance: All ionogels—regardless of IL hydrophobicity—maintained frost-free transparency and durability at -70°C. For instance, even highly hygroscopic ILs like [Emim]Cl (which is solid at room temperature but forms stable ionogels) showed no crystallization or opacity under deep cooling.

Thermoresponsive behavior: While hydrophobicity influenced thermoresponsiveness (e.g., only [Emim][TFSI]/BA and [Emim][BF₄]/HEA ionogels exhibited thermosensitive phase transitions, the former is LCST, and the latter is UCST), it did not dictate low-temperature stability. Hygroscopic ILs produced ionogels that were equally stable against freezing, though they lacked thermosensitive properties.

Thus, hydrophobicity and low volatility are not necessary conditions for low-temperature durability and frost-free transparency. Instead, the key factor is the IL's intrinsic ability to resist crystallization due to molecular asymmetry, which is a universal characteristic of most ILs. This generality allows for a wide range of IL selections based on application needs without compromising low-temperature performance. We have expanded the discussion in the manuscript to incorporate these insights and thank the reviewer for prompting this deeper analysis. Our findings highlight the versatility of IL-based ionogels for applications in extreme environments.

“Supplementary Note 3. Hydrophobicity and low-temperature resistance of common imidazolium ILs.

*To investigate the correlation between hydrophobicity, volatility, melting point, and low-temperature tolerance of ILs, four ILs sharing the common 1-ethyl-3-methylimidazolium ([Emim]⁺) cation but differing anions were selected: [Emim][TFSI], [Emim][BF₄], [Emim]Cl, and [Emim][EtSO₄] (**Supplementary Fig. 22a**). The anion variation imparts distinct hydrophobicity, hygroscopicity, and thermodynamic properties. Their melting points are approximately -18 °C, 15 °C, 84 °C, and -37 °C, respectively. All exhibit negligible vapor pressure at elevated temperatures. At room temperature, [Emim]Cl is solid and highly hygroscopic due to its chloride anion (**Supplementary Fig. 22b**). Hydrophilicity follows the order: [Emim][TFSI] < [Emim][BF₄] < [Emim]Cl < [Emim][EtSO₄], while hydrophobicity shows the opposite*

trend. Hydrophobicity was confirmed by water miscibility: [Emim][TFSI] is completely immiscible (most hydrophobic), whereas the other three are miscible (**Supplementary Fig. 22c**).

For monomer selection, the hydrophobic monomer butyl acrylate (BA) is compatible with hydrophobic [Emim][TFSI], while the more hydrophilic ionic liquids ([Emim][BF₄], [Emim]Cl, [Emim][EtSO₄]) show better compatibility with the hydrophilic monomer 2-hydroxyethyl acrylate (HEA) (**Supplementary Fig. 21d**). The crosslinker and photoinitiator were consistent with those used in this work. After the same low-temperature UV curing process, four distinct ionogels were obtained (**Supplementary Fig. 22e**). Similar to the high-performing [Emim][TFSI]/BA system, the [Emim][BF₄]/HEA combination exhibited reversible upper critical solution temperature (UCST) behavior, turning transparent at high temperatures and opaque at low temperatures. In contrast, ionogels based on [Emim]Cl and [Emim][EtSO₄] showed no observable thermosensitive phase transition. Remarkably, after identical low-temperature treatment, all four ionogels demonstrated excellent low-temperature tolerance while fully retaining their respective phase transition characteristics (**Supplementary Fig. 22f**). Even at the experimental lower limit of -70 °C, all ionogels maintained excellent performance.

This low-temperature tolerance is primarily attributed to the inherent properties of imidazolium ILs. Their structural asymmetry, arising from bulky, asymmetric ions, inhibits regular crystalline packing upon cooling, suppressing crystallization and leading to low melting points and a propensity for supercooling. Furthermore, physical interactions (e.g., ion-dipole) between the ILs and polymer chains are believed to further inhibit IL crystallization, thereby enhancing the low-temperature tolerance of the ionogels¹⁶.

Supplementary Figure 22. Comparison of hydrophobicity and low-temperature tolerance among four commonly used ILs. a) The structures of the anions and cations of four ILs, where the cations are identical. b) Optical photographs of ILs. c) Demonstration of miscibility with water; IL:water = 1:1 wt/wt. d) Demonstration of precursor uniformity; IL:monomer = 1:1 wt/wt. e) Optical demonstration of ionogels. f) Optical demonstration of ionogels after cryogenic freezing at -70°C (achieved using a dry ice-ethanol bath). ”

4. The device reads as mechanically robust and therefore promising; nevertheless, optical components must withstand harsh environments and strong UV exposure in real service. I recommend adding accelerated aging test like high temperature–humidity, UV-A/B exposure, or salt-fog/corrosive atmospheres.

Response: We sincerely thank the reviewer for this valuable suggestion regarding the long-term durability of ATI under harsh environmental conditions. We fully agree that assessing stability against factors like strong UV exposure, temperature-humidity cycles, and corrosive atmospheres is crucial for evaluating its real-world application potential. In direct response to this comment, we have conducted a comprehensive set of accelerated aging tests to rigorously evaluate the environmental robustness of the ATI. Specifically, we performed: i) High temperature and high humidity cycle tests (4 h at 60°C followed by 4 h at >90% RH, for 5 cycles); ii) Salt spray aging tests using a 5 wt% sodium chloride aqueous solution (pH = 7) in a high-humidity environment (>90% RH) for 24 hours, consistent with methodologies outlined in relevant standards for assessing corrosion resistance; and iii) Ultraviolet tests involving separate exposures to UVA (365 nm, 10 mW cm⁻²) and UVB (311 nm, 1 mW cm⁻²) radiation for 12 hours.

The results, now included in the manuscript as **Supplementary Fig. 27, 28, and 29**, demonstrate the outstanding robustness of the ATI. A particularly noteworthy finding is that while intense UV exposure induces temporary photo-yellowing, this discoloration significantly recedes after storage in ambient air. Most importantly, the core thermochromic functionality and excellent low-temperature tolerance of the material remain almost unaffected.

“The stability of ATI was further evaluated under a series of complex accelerated aging protocols, including cyclic high-temperature/humidity conditions, salt spray tests, and UVA/UVB radiation. The experimental results confirm its outstanding robustness (Supplementary Fig. 27 and 28). It is noteworthy that while intense UV exposure leads to some photo-induced yellowing, this phenomenon is temporary. The discoloration significantly recedes after storage in ambient air, and the thermochromic functionality and low-temperature tolerance remain almost unaffected (Supplementary Fig. 29).”

Supplementary Figure 27. High temperature and high humidity cyclic testing for ATI.
 a) High temperature and high humidity cyclic testing process. b) ATI optical photographs after each cycle, totaling 5 cycles.

Supplementary Figure 28. Salt spray testing for the ATI. a) On-site photo of salt spray testing. b-f) Thermochromic performance of ATI after 24 h salt spray testing.

Supplementary Figure 29. Long-term stability of ATI under UVA and UVB irradiation, and FTIR spectra before and after irradiation.

Although ATI currently demonstrates considerable environmental tolerance, there remains room for improvement to meet the future demands for extreme environmental resilience in smart materials. Therefore, in the concluding discussion section, we present prospects for ongoing research into ATI's environmental tolerance.

“To enhance the application potential of ATI in future, the following two aspects should be further investigated: i) optimizing the forming technology of asymmetric structures to mitigate constraints imposed by mold area and intricate preparation processes on the large-scale production of ATI; and ii) leveraging the modifiability of polymers to continuously improve the environmental robustness of ATI, particularly under complex extreme conditions such as durability at ultra-high temperatures (exceeding 100°C), toughness at ultra-low temperatures (below liquid nitrogen temperature), and performance in multifaceted harsh environments (e.g., under strong UV irradiation coupled with high temperature and humidity).”

We believe these newly added experiments and discussions directly address the reviewer's concern by providing concrete evidence of the ATI's resilience under simulated harsh conditions, thereby strengthening the case for its practical applicability. We are grateful for this insightful suggestion, which has enhanced the depth of our stability analysis.

5. To illustrate practical impact, the measured optical states should be coupled to building-scale energy simulations under representative climates.

Response: We sincerely thank the reviewer for this excellent suggestion, which is crucial for demonstrating the practical energy-saving potential of our ATI smart windows in real-world building applications. We fully agree that linking the measured optical properties to building energy performance across diverse climates can significantly strengthen the impact of our work.

In direct response to this comment, we have conducted a comprehensive set of building energy simulations using EnergyPlus. The simulations were designed to quantitatively evaluate the annual energy-saving benefits of our ATI smart windows in different representative Chinese climates. The revisions and the methods used are as follows:

“Energy-saving performance simulation

*To evaluate the energy-saving and cooling performance of the ATI smart window, we conducted building energy simulations using the open-source software EnergyPlus (version 23.2.0). The building model was constructed using SketchUp (version Pro 2021). A typical container house model measuring 27 m (L) × 18 m (W) × 4 m (H) (6 stories) was employed (**Supplementary Fig. 46**). Through whole-building energy*

simulation, the HVAC energy savings attributable to the test sample were estimated. First, a baseline scenario was established without the ATI smart window. The HVAC system was configured to activate heating mode when the ambient temperature dropped below 16 °C, and cooling mode when the indoor temperature exceeded 26 °C. The corresponding HVAC energy consumption was computed under these conditions. Subsequently, the ATI was incorporated into the model as a window energy-saving component, maintaining the original building structure. The HVAC energy consumption was recalculated under identical operational settings. Climate data for various cities, obtained from the EnergyPlus website (<https://energyplus.net/weather>), were used to simulate the HVAC energy demands for both the baseline and ATI-integrated building models across different climatic conditions.

Supplementary Figure 46. Schematic of 27 m × 18 m × 4 m (6 stories) simplified houses for EnergyPlus simulation.

Results

To comprehensively illustrate the energy-saving capabilities of ATI smart windows, we conducted an extensive simulation with EnergyPlus to evaluate the annual energy consumption of a building model fitted with ATI smart windows (Supplementary Fig. 41). Compared to clear windows with pure glasses, ATI smart windows deliver more significant energy savings in China's warmer urban areas, highlighting their potential for application in solar-regulated devices (Fig. 5h).

Fig. 5: Passive smart cooling windows realized by ATI. *a* Proposed light modulation schematic for ATI smart windows. *b* Hydrophobicity of ATI smart windows, including contact angle (i) and adhesion (ii) of ATI-BN compared with glass and PNIPAM hydrogel, as well as simulated raindrop sliding test (iii). *c* Demonstration of autonomous adhesion of ATIs on different common transparent substrates. *d* UV-Vis-NIR transmittance spectra of ATI-based windows at 20–45°C, together with the solar irradiance intensity (air mass 1.5 global) and photopic luminous efficiency. *e* The temperature variation of model rooms with different windows under the illumination of simulated sunlight (xenon lamp) over time; inset is an optical photograph of the test site. *f* On-site optical photos of the model rooms during the cooling performance testing process. *g* Variation of room temperature in model rooms with different windows within 24 h (Xi'an, 34°15'N, 108°40'E, May 20, 2025). *h* China energy-saving map of the building model with ATI smart windows.

Supplementary Figure 41. Annual energy savings achieved by the ATI smart window model house in provincial capital cities across China through Energy Plus simulation. ”

We believe that these building-scale simulation results, made possible by the reviewer's insightful suggestion, provide strong and direct evidence of the practical energy-saving impact of our ATI smart windows across diverse geographical conditions. This addition greatly enhances the reference value and broad interest of our manuscript.

6. The current SAXS analysis is qualitative; please perform quantitative structure-factor fitting to extract real-space correlation lengths or domain sizes and present their temperature-dependent evolution alongside scattering intensity.

Response: We sincerely thank the reviewer for this insightful suggestion. We agree that a quantitative analysis of the scattering data will provide deeper insights into the structural evolution of the material. In direct response to this comment, we have performed a quantitative analysis of the WAXS data.

Specifically, we have supplemented the variable-temperature WAXS profiles (including 1D spectra and 2D patterns) for the ATI-B sample at 20°C, 30°C, and 40°C. The characteristic real-space dimensions (correlation lengths) corresponding to the scattering peaks were extracted using the characteristic length calculation formula (Formula 1 in the revised manuscript). The quantitative results are now summarized in the new Supplementary table 1 and discussed in the main text. The key quantitative finding, as added to the manuscript, is as follows:

“The variable-temperature WAXS results of ATI-B reveal that the characteristic dimensions of its scattering peaks shift from 4.5 Å and 7.7 Å to 4.6 Å and 7.2 Å (Fig. 2d-ii, Supplementary Fig. 14 and Table 1). This change can be attributed to the increase in polymer chain spacing due to thermal expansion effects, coupled with the entropy-driven contraction of [Emim]⁺-[TFSI]⁻ ion clusters⁴¹”

Fig. 2: Preparation, composition, and characterization of asymmetric structures of ATIs. *a* ESP simulations of key ATI components: polymer monomers BA and NIPAM, and IL solvent [Emim][TFSI]. *b* FTIR spectra of solvent-free PBA, pure IL, and ATI-B. *c* Microscopic image (left) and SEM image (right) of the cross-section of the ATI. *d* 2D spectra of SAXS (i) and WAXS (ii). *e* Raman spectra of ATI-B and ATI-BN; insets i, ii, iii are Raman mapping images of ATI-B, ATI-BN, and ATI cross sections, respectively. *f* Optical images of physical bilayer ionogel and ATI (upper PBA ionogel layer, stained with metanil yellow; lower P(BA-co-NIPAM) ionogel layer, stained with brilliant green) and schematic of the proposed interface. *g* Characterization of the rheological properties of ATIs, including amplitude sweeping (left), frequency sweeping (middle), and temperature ramping (right).

Supplementary Figure 14. 1D WAXS spectra of ATI-B.

Supplementary Table 1. d-value in the variable-temperature WAXS of ATI-B.

	20°C	30°C	40°C
d-1	7.7 Å	7.6 Å	7.2 Å
d-2	4.5 Å	4.5 Å	4.6 Å

7. Please undertake a careful language and figure audit. Avoid typos like, in Fig. 5d the axis label should be labeled “wavelength.”

Response: We sincerely thank the reviewer for highlighting this oversight and for the suggestion to perform a thorough audit of the manuscript. We deeply appreciate the reviewer's meticulous attention to detail. In direct response to this comment, we have conducted a comprehensive, line-by-line language and figure audit of the entire manuscript. Specifically, the axis label in Fig. 5d has been corrected to "Wavelength". Furthermore, we have meticulously re-examined all text, figure labels, axis titles, unit designations, and grammatical structures to identify and rectify any similar typographical or formatting inconsistencies. We are confident that this rigorous audit has greatly enhanced the precision and professionalism of the manuscript, and we are grateful for this suggestion.

Reviewer #3 (Remarks to the Author)

The manuscript by Du et al. presents a technically impressive and well-executed study on a bio-inspired, asymmetric thermochromic ionogel (ATI). The authors have successfully engineered a material with an outstanding combination of mechanical toughness, environmental resilience (especially low-temperature performance), and dual-mode optical switching capabilities. The experimental work is thorough, spanning multi-scale characterization and theoretical simulations. Overall, this work is highly innovative and significant, and its findings will be of broad interest to the fields of smart materials, flexible electronics, and energy-saving buildings. I recommend that this manuscript be accepted for publication in Nature Communications after the authors address the following minor points.

Response: We are deeply grateful to the reviewers for the time and effort dedicated to evaluating our manuscript, and for providing positive and constructive feedback. We sincerely appreciate the reviewer's generous assessment that our work is "highly innovative and significant," presenting a "technically impressive and well-executed study" with findings of "broad interest." Such encouraging comments are a tremendous motivation for our team. We are particularly pleased that the reviewer recognized the outstanding combination of mechanical toughness, environmental resilience, and dual-mode optical switching capabilities of the developed asymmetric thermochromic ionogel (ATI). In accordance with the reviewer's insightful suggestions, we have carefully addressed each of the minor points raised. We believe that the revisions and clarifications made have further enhanced the clarity and robustness of the manuscript. A point-by-point response to all comments is provided below.

1. In the main text (Fig. 3), the authors use MD simulations to show that an increase in temperature weakens the interaction energy between PBA and the ions while significantly strengthening the ion-ion interactions. This is a central argument. It is suggested that this point be more explicitly linked in the main text to the peak shifts observed in the FTIR and Raman spectra. This would create a stronger connection between theory and experiment and more directly lead to the conclusion of "entropy-driven phase separation." For example: "The temperature-dependent FTIR results (Fig. 3e) indicate a weakening of the ion-dipole interaction, an observation strongly corroborated by our MD simulations (Fig. 3h), which calculate a significant decrease in the PBA-ion interaction energy. Likewise, the TDRS spectra (Fig. 3f) suggest an enhancement of the ion-ion Coulomb forces, a phenomenon perfectly mirrored in our simulations where the

[Emim]⁺-[TFSI]⁻ interaction becomes dominant. Taken together, these tightly coupled experimental and theoretical results provide compelling evidence for an entropy-driven phase separation mechanism.”

Response: We sincerely thank the reviewer for this excellent suggestion, which indeed strengthens the connection between our experimental spectroscopic evidence and the theoretical simulations. We agree that making this link more explicit significantly enhances the fluency of the narrative and provides a more direct and compelling argument for the entropy-driven phase separation mechanism. Following the reviewer's insightful advice, we have revised the main text in the Results section (specifically in the subsection discussing Figure 3) to explicitly integrate the FTIR/Raman spectral shifts with the MD-calculated interaction energies, as suggested. The added text closely mirrors the example provided by reviewer for clarity and impact:

“The microscopic interactions in thermosensitive dynamic phase separation were analyzed by in situ characterization and computational chemistry techniques. The temperature-dependent FTIR results (Fig. 3e) indicate a redshift of the characteristic peaks of ATI-B (from 1052 to 1050 cm⁻¹ and 1348 to 1346 cm⁻¹), suggesting a weakening of the ion–dipole interaction between the IL and polymer⁵¹. This observation is strongly corroborated by density functional theory (DFT) calculation, which shows a significant decrease in the PBA-ion interaction energy per unit area (Supplementary Fig. 30). Likewise, the temperature-dependent Raman spectroscopy (TDRS) reveals a shift of the ν(-N-S) peak (Fig. 3f, Supplementary Fig. 31) to lower wavenumbers, implying enhanced Coulomb interaction between [Emim]⁺ and [TFSI]⁻⁵². This phenomenon is perfectly reflected in the molecular dynamics (MD) simulation, where the [Emim]⁺-[TFSI]⁻ interaction energy increases with rising temperature and becomes dominant (Fig. 3g). Furthermore, two-dimensional correlation infrared spectroscopy (2DCOS FTIR) analysis suggests that the phase separation is triggered by the preferential dissociation of anions from the polymer, followed by cation re-association (Supplementary Fig. 32, Table 3 and Note 4). Taken together, these tightly coupled experimental and theoretical results provide compelling evidence for an entropy-driven dynamic phase separation mechanism above the LCST, wherein enhanced Coulomb forces promote the aggregation of solvent molecules detached from the polymer (Supplementary Note 5)⁵³.

Fig. 3: Dynamic phase separation principle and microscopic interaction analysis of ATIs. *a* Proposed schematic of the thermosensitive dynamic phase separation in ATI-B. *b* Optical microscopy images of ATI during the heating process. *c* Proposed schematic of the propagation path of light inside ATI-B below (left) and above (right) the LCST. *d* Optical photos of ATI under different temperatures. *e* Temperature-dependent FTIR spectra. *f* Temperature-dependent Raman spectra (colour scale: red, high intensity; blue, low intensity). *g* MD snapshots of PBA and [Emim][TFSI] at 298K and 313K. The calculated interaction energies of ion-dipole interactions (PBA-[Emim]⁺, PBA-[TFSI]⁻) and ion-ion interactions ([Emim]⁺-[TFSI]⁻) in ATI at 298K and 313K; insets show the ESP of the optimized structure of [Emim]⁺-[TFSI]⁻ at the two temperatures (colour scale: red, negative; blue, positive).”

We believe this revision creates a much stronger synergy between theory and experiment, as intended by the reviewer, and we are grateful for this valuable input that has improved the clarity and impact of our central argument.

2. The manuscript demonstrates excellent stability over 100 heating-cooling cycles (Supplementary Fig. 20) and stability after 24 hours of continuous heating (Supplementary Fig. 21). While this is sufficient for a laboratory evaluation, real-world applications like smart windows require the material to endure much longer periods (several years) of UV irradiation and temperature fluctuations. The authors are advised to add a brief perspective or preliminary assessment in the discussion section regarding the material's long-term chemical stability under UV irradiation.

Response: We sincerely thank the reviewer for this insightful comment regarding the long-term durability of ATI under real-world conditions, particularly concerning UV irradiation and complex environmental fluctuations. We completely agree that assessing long-term stability beyond standard laboratory cycling is crucial for evaluating the material's potential in applications like smart windows. In direct response to this suggestion, we have conducted a series of accelerated aging tests to preliminarily evaluate the material's long-term chemical stability and durability. Specifically, we performed:

- i) High-temperature and high-humidity cycle testing
- ii) Salt spray aging test
- iii) UV irradiation tests, including UVA and UVB

The results of these tests consistently demonstrate that our ATI material retains its core thermochromic functionality and structural integrity without significant performance degradation. These new findings have been added as **Supplementary Fig. 27, 28, and 29**, respectively. Furthermore, as suggested by the reviewer, we have incorporated a brief perspective and a discussion of these preliminary accelerated aging results in the main text's Results section. The added text reads:

“For LCST-type TCPs, the sensitivity and long-term resistance to temperature changes directly affect their performance in practical application⁴⁹. ATI exhibits rapid response and significant optical changes above and below LCST (Fig. 3d). It is noteworthy that, owing to the intrinsic nonvolatile nature and amorphous structure of the ILs, along with the suppression of IL crystallization through ion–dipole interactions⁵⁰, ATI remains transparent without undergoing an opaque transition caused by ice crystal formation, even after prolonged exposure to a low temperature of

–70 °C (**Supplementary Fig. 22, Note 3, and Movie 3**). Even after prolonged exposure to low temperatures, ATI undergoes opacity switching in a short period when returning to temperatures above LCST (**Supplementary Fig. 23**). In contrast, the growth of ice crystals in PNIPAM hydrogels leads to the destruction of the integrity of the polymer network after multiple freeze-thaw cycles (**Supplementary Fig. 24**). In 100 cycles of testing from –45°C to 40°C, the transmittance of ATI to 550 nm wavelength light does not undergo significant degradation (**Supplementary Fig. 25**). Moreover, ATI does not experience significant dimensional shrinkage or functional failure even when subjected to long-term operating conditions above LCST (**Supplementary Fig. 26**). The above advantages enable ATI to effectively circumvent the limitations of low-temperature freezing and high-temperature water loss faced by hydrogel-based TCPs. The stability of ATI was further evaluated under a series of complex accelerated aging protocols, including cyclic high-temperature/humidity conditions, salt spray tests, and UVA/UVB radiation. The experimental results confirm its outstanding robustness (**Supplementary Fig. 27 and 28**). It is noteworthy that while intense UV exposure leads to some photo-induced yellowing, this phenomenon is temporary. The discoloration significantly recedes after storage in ambient air, and the thermochromic functionality and low-temperature tolerance remain almost unaffected (**Supplementary Fig. 29**).

Supplementary Figure 27. High temperature and high humidity cyclic testing for ATI. a) High temperature and high humidity cyclic testing process. b) ATI optical photographs after each cycle, totaling 5 cycles.

Supplementary Figure 28. Salt spray testing for the ATI. a) On-site photo of salt spray testing. b-f) Thermochromic performance of ATI after 24 h salt spray testing.

Supplementary Figure 29. Long-term stability of ATI under UVA and UVB irradiation, and FTIR spectra before and after irradiation.

We believe these additions directly address the reviewer's concern by providing experimental evidence and a reasoned perspective on the material's long-term stability,

thereby significantly strengthening the discussion regarding the ATI's potential for real-world applications. Once again, we are grateful for this valuable suggestion.

3. In the active optical display section (Fig. 6), the authors demonstrate both direct and indirect Joule heating. The main text or figure captions should explicitly state the voltage, current, or power density required to achieve the corresponding temperatures. For example, the SI mentions that a voltage of 180V and a frequency of 10 kHz are used to achieve global opacity.

Response: We thank the reviewer for this precise comment, which is crucial for readers to accurately reproduce and understand the electro-thermal driving conditions of our active optical display. We agree that explicitly stating the electrical parameters will significantly enhance the clarity and practical value of **Figure 6**.

In direct response to this suggestion, we have now revised the captions for **Figure 6e** to explicitly include the applied voltage (V) and frequency (kHz) used to achieve the specific heating effects and corresponding temperatures shown in the display demonstrations.

Fig. 6: Flexible dynamic optical display realized by ATI. *a* Schematic of a flexible dynamic display device driven by localized indirect heating. *b* Optical photographs of a localized optical display device. *c* Demonstration of localized optical displays, including optical photographs and thermal infrared images in the planar state (i), and localized optical displays on curved substrates (ii). The DC voltage for the conductive Joule heating layer is 24 V. *d* Schematic and equivalent circuit diagram of the ATI-based global Joule heating system. *e* Demonstration of in situ Joule heating of ATI using an AC power supply (180 V, 10 kHz), including optical and thermal infrared images before and after power on in the planar state (i), as well as thermal infrared images of ATI in the stretched and bent states when power on (ii). *f* Schematic of a smart projection display system based on ATI optical display switching; and *g* Realistic demonstration of the ATI smart projection display system, including the global projection of the famous painting of “Dance” and the traditional Chinese calligraphy “Ascending to the Height”, as well as localized magnified images (inset). *h* Demonstration of a dynamic localized projection display. The upper half is an opaque area due to localized heating for displaying the weather conditions of the day, and the lower half is a highly transparent due to not being heated.

We believe these additions to the figure captions fully address the reviewer's point by providing the essential experimental parameters directly alongside the results, making the manuscript more complete and reproducible.

4. To broaden the context and impact of the manuscript, the authors might consider expanding the Introduction to include more foundational literature on ionogel science. While the review of smart windows is relevant, acknowledging the pioneering work on thermo-sensitive gels by Yongjun Men (Donghua University), high-strength ionogels by Feng Yan (Soochow University), and related supramolecular systems by Minghua Liu (Beihang University) would provide a more complete scientific background for the achievements presented here.

Response: We sincerely thank the reviewer for this excellent suggestion, which has guided us to significantly strengthen the scholarly foundation of our manuscript. We agree that anchoring our work within the broader landscape of key advances in ionogel science provides a more complete and authoritative background for highlighting our achievements.

As recommended, we have expanded the Introduction section by adding a new paragraph that thoughtfully integrates the foundational work of the mentioned research groups. The added text now includes:

Acknowledges the contributions of Prof. Feng Yan's group to the development of high-strength and high-toughness ionogels with mechanical performance surpassing that of metals and alloys (**references 28, 29**), establishing a solid foundation for robust ionogel-based thermochromic polymers (TCPs).

Incorporates the comprehensive reviews by Prof. Yongjun Men's group on LCST- and UCST-type thermosensitive ionogels (**references 30, 31**), which expertly summarize the progress in addressing the limitations of conventional gels.

Connects these advances to the core theme of our work by discussing how the intrinsic properties of ionic liquids (ILs) enable stable thermoresponsive behavior under complex conditions, citing an example of IL-based thermochromic ionogels (**reference 33**).

Furthermore, this integration allows us to clearly position our contribution: after reviewing these significant achievements, the added text concludes by identifying the existing challenges—such as complex synthesis and high costs—that currently hinder large-scale commercialization. This lays a perfect foundation for introducing ATI as a novel approach aimed at overcoming these very limitations through its unique design. The details of the revision are as follows:

“ In recent years, Yan and colleagues have systematically optimized the supramolecular structures of IL-based functional materials, leading to the development of a series of high-strength and high-toughness ionogels. Remarkably, the mechanical

performance of these ionogels surpasses that of most metals and alloys^{28, 29}, providing a solid foundation for fabricating mechanically robust ILs-based TCPs. Comprehensive reviews by Men et al. summarize the progress in LCST- and UCST-type thermosensitive ionogels, highlighting their role in addressing the limitations of conventional hydrogels and organogels^{30, 31}. Owing to the low glass transition temperatures and tunable supramolecular characteristics of ILs³², TCP-based ionogels are expected to maintain stable thermoresponsive behavior under complex and variable temperature conditions. For instance, IL-based thermochromic ionogels fabricated using modified PU/polyvinylidene fluoride (PVDF) matrices exhibit a tunable LCST between 20 to 40 °C, combined with robust mechanical properties and improved thermal stability³³. However, challenges such as complex synthesis routes, stringent processing conditions, and high production costs currently hinder their large-scale commercialization. Therefore, facile fabrication strategies for high-performance TCPs remain a significant challenge, necessitating deeper understanding of the dynamic phase separation mechanisms governing IL-polymer network interactions—a fundamental prerequisite for optimizing ionogel-based TCPs and advancing next-generation smart thermal-responsive systems. ”

We are confident that this revision has successfully broadened the scientific context of our manuscript. We believe the Introduction is now substantially more comprehensive and impactful.

Point-by-Point Response to Reviewers' Comments

Manuscript ID: NCOMMS-25-68161B

(Comments in black, responses in **light blue**, revised texts in *blue italics*)

We are deeply grateful to the editor and reviewers for their thorough review and valuable comments, which have greatly improved our work. Our point-by-point responses are provided below. We sincerely appreciate the time and expertise you have dedicated to our manuscript.

Reviewer #1 (Remarks to the Author)

The authors have made extensive revisions by addressing all the comments and provide additional experimental results. Now the statements and conclusions are well supported by experimental data and proper analysis. It is recommended for publication in Nature Communications in its current form.

Response: Thank you for your kind words and positive assessment of our revisions. We are very pleased that you find the manuscript now suitable for publication. We sincerely appreciate your time and insightful guidance throughout the review process, which have been valuable in strengthening our work.

[Editor's Note: This Reviewer was asked to evaluate the response to Reviewer #2's comments, and provided the following:]

The authors have tried to respond to all the previous comments from Reviewer #2. However, some critical issues remain.

Response: Thank you for your continued evaluation and for highlighting that some critical issues require further attention. We have carefully considered all points raised and are fully committed to addressing all remaining concerns. Below is the detailed point-by-point response:

1. The SAXS and WAXS results indicate a phase separation domain size of about 7.2 nm, and some unspecified characteristic dimension of 4.5 Å and 7.2 Å. These dimensions could not result in any severe scattering of visible-NIR light that causes a transparency-to-opaque transition.

Response: Thank you for your professional review and attention to this manuscript. We apologize for any confusion caused by inaccurate expressions. The following provides further clarification regarding X-ray scattering characterization:

i) About SAXS:

Indeed, to demonstrate the in-situ phase-separated structure within the mechanical support layer ATI-BN, we performed SAXS characterization to obtain its internal phase-separated dimensions (Supplementary Figure 13). This enabled us to explain its mechanical toughening effect through the interfacial load transfer mechanism. The proposed in-situ phase-separated toughening principle is illustrated as Supplementary Figure 34:

Supplementary Figure 13. Analysis of bicontinuous structure in ATI-BN. a) 1D SAXS spectra. b) SEM image at the interface of ATI, with ATI-BN (rough) on the left and ATI-B (smooth) on the right. c) The proposed schematic diagram of the bicontinuous structure and phase-separated correlation length in ATI-BN.

Supplementary Figure 34. Schematic of interphase load-transfer and energy-dissipation pathway in a phase-separated structure.

In-situ phase separation is a classical mechanical toughening technique for soft matter that has been extensively validated and tested. The detailed toughening mechanism can

be found in the following literature: Tough and stretchable ionogels by in situ phase separation. *Nat. Mater.* 2022, 21, 359–365.

ii) About WAXS:

WAXS characterization was obtained based on the LCST ionogel layer ATI-B. In fact, we can directly observe the macroscopic phase separation process of ATI-B above its LCST using optical microscopy. This process occurs at the micrometer scale, allowing direct visualization of the formation, aggregation, and expansion of ionic liquid domains (**Figure 3b**). It must be acknowledged that real-time characterization of the phase separation feature sizes within ATI-B at critical temperature nodes is an extremely challenging task. We have made numerous attempts and exerted considerable effort for this, yet not yielded satisfactory results. Despite this, we still sought to investigate the molecular-scale dynamic changes in ATI-B during heating to indirectly understand its LCST behavior. For this purpose, we used WAXS to observe the characteristic dimensions of ions during heating, as this technique is suitable for characterizing features between 0.1 and 10 nm—precisely the size range of anions and cations in ionic liquids. These efforts aim to provide readers with a deeper understanding of LCST behavior at the ionic scale, specifically the initial formation and expansion of ionic clusters. Furthermore, the extensive formation of these ionic clusters will lead to macroscopic aggregation of ionic liquid domains, thereby affecting the gel's transmittance via light scattering. In summary, although the characteristic size obtained by WAXS is far below the range required to affect the light scattering performance of ion gel, it can help us understand the characteristic size of ion clusters at the initial stage of phase separation. We hope these explanation will address your concerns and earn your approval.

Supplementary Figure 14. 1D WAXS spectra of ATI-B.

Supplementary Table 1. *d*-value in the variable-temperature WAXS of ATI-B.

	20°C	30°C	40°C
d-1	7.7 Å	7.6 Å	7.2 Å
d-2	4.5 Å	4.5 Å	4.6 Å

Fig. 3: Dynamic phase separation principle and microscopic interaction analysis of ATIs. *a* Proposed schematic of the thermosensitive dynamic phase separation in ATI-B. *b* Optical microscopy images of ATI during the heating process. *c* Proposed schematic of the propagation path of light inside ATI-B below (left) and above (right) the LCST. *d* Optical photos of ATI under different temperatures. *e* Temperature-dependent FTIR spectra. *f* Temperature-dependent Raman spectra (colour scale: red, high intensity; blue, low intensity). *g* MD snapshots of PBA and [Emim][TFSI] at 298K and 313K. The calculated interaction energies of ion-dipole interactions (PBA-[Emim]⁺, PBA-[TFSI]⁻) and ion-ion interactions ([Emim]⁺-[TFSI]⁻) in ATI at 298K and 313K; insets show the ESP of the optimized structure of [Emim]⁺-[TFSI]⁻ at the two temperatures (colour scale: red, negative; blue, positive).

2. On the other hand, the authors' explanation to the dynamic phase separation is based on the ion-dipole interaction at low temperatures and its transition to ion-ion transition at high temperatures, which may violate some basic physics. Once again, the origin of LCST of the ionogels is not well established.

Response: We sincerely thank the reviewers for their time and expertise in evaluating our manuscript. We appreciate your thoughtful comments regarding the transition of ion-dipole interactions and the origin of the LCST. In our previous response and revisions, we have tried to address this point and clarify any potential misunderstandings. To further support the credibility of mechanism discussed, we have reviewed relevant discussions on this mechanism from influential research teams in the field and excerpted them, with key points highlighted in **bold** for easy reference. We hope to receive your approval and endorsement. Thank you once again for any effort you have put into this manuscript.

i) Chao Wang Group at Tsinghua University

A Stable and Self-Healing Thermochromic Polymer Coating for All Weather Thermal Regulation. Adv. Funct. Mater. 2023, 33, 2307240.

“We think under high temperature, the ion–dipole interaction is weakened so the force between the cation and CF₃ dipoles is not as strong as that at low temperature, leading to the aggregation of ionic liquid. To confirm our hypothesis, we compare the Raman spectra of TPC film at transparent and opaque states (Figure 2h). The band of stretching vibrations of N–S bonds in [TFSI] anion at around 742 cm⁻¹ can be attributed as an indicator of ionic association. Increasing the temperature causes the peak at 744 cm⁻¹ shift toward lower wavenumbers (742 cm⁻¹). The redshift indicates N–S bonds are weaker. **This is because the Coulombic force between [TFSI] anion and [dhimi] cation is strengthened, leading to less negative charges on the N atom of [TFSI] anions. The stronger ion–ion interaction causes the aggregation of IL, forming a homogeneous opaque film.** The DSC results (Figure 2i) show there is a small enthalpy change around 43 °C, indicating the phase separation of the material. The low ΔH implies the TPC film has a well dimensional stability because a large enthalpy change always implicates a large-scale phase separation coinciding with liquid expulsion. From a thermodynamic point of view, a negative enthalpy of mixing (ΔH_{mix} < 0) and a negative entropy of mixing (ΔS_{mix} < 0) are essential for LCST phase behavior. Many works reported the hydrogen-bonding interaction or cation–π interaction can both lead to the formation of ordered structures between IL and polymer so caused a negative ΔS_{mix}. Similarly, we speculate the oriented solvation between the polymer and ionic liquid caused by ion–dipole interactions in our system may serve as the driving force

for LCST phase behavior (Figure 2j). At room temperature, the Gibbs free energy of mixing ($\Delta G_{\text{mix}} = \Delta H_{\text{mix}} - T\Delta S_{\text{mix}}$) is negative. The endothermic breaking of ion–dipole interaction makes ΔH_{mix} less negative and the entropy gain leads to the increase in the entropy term ($|T\Delta S_{\text{mix}}|$). Once $|T\Delta S_{\text{mix}}|$ exceeds $|\Delta H_{\text{mix}}|$, ΔG_{mix} becomes positive and the entropy-driven phase separation occurs, wherein ILs aggregate into isolated domains.”

ii) Yang Li Group at Jilin University

Self-Adhesive Self-Healing Thermochromic Ionogels for Smart Windows with Excellent Environmental and Mechanical Stability, Solar Modulation, and Antifogging Capabilities. *Adv. Mater.* 2023, 35, 2211456.

“In a gel system, the LCST phenomenon is attributed to the reversible interaction between the polymer and solvent, which determines the solvation condition of the polymer in the solvent.

.....

The underlying reason for the reversible thermochromic function is scrutinized by variable-temperature FTIR spectroscopy and optical microscopy and is schematically shown in Figure 1b. Figure 1d(1) shows no phase separation occurring inside the SPU_{1.2}/30%IL_{0.5} ionogel at room temperature. By comparing the FTIR spectra of binary ILs and the SPU_{1.2}/30%IL_{0.5} ionogel at 20 °C (Figure 1e), it can be seen that the presence of SPU_x causes the C-H stretching peaks of the imidazolium ring at 3154 cm⁻¹ and 3111 cm⁻¹ shift toward higher wavenumbers, confirming the formation of hydrogen bonds between the [Bdmim][TFSI]/[Bmim][TFSI] and ether groups in PPG segments (Figure 1f). The formation of hydrogen bonds is an exothermic process, which results in a negative enthalpy change ($\Delta H_{\text{mix}} < 0$). Moreover, the formation of hydrogen bonds leads to a negative entropy ($\Delta S_{\text{mix}} < 0$) due to the formation of an organized IL molecule layer around the PPG segments. At room temperature, the Gibbs free energy of mixing ($\Delta G_{\text{mix}} = \Delta H_{\text{mix}} - T\Delta S_{\text{mix}}$) is negative as $|\Delta H_{\text{mix}}|$ is larger than $|T\Delta S_{\text{mix}}|$. Thus, ILs are evenly dispersed in the polymer network, ensuring good transparency. The FTIR spectra of the SPU_{1.2}/30%IL_{0.5} ionogel at 20, 40, and 60 °C were presented in Figure 1e. Results show that increasing the temperature above the cloud point temperature (T_{cp}) causes the peaks at 3155 and 3115 cm⁻¹ to shift toward lower wavenumbers, indicating the cleavage of hydrogen bonds between SPU_x and ILs. This was further confirmed by variable-temperature ¹H NMR spectroscopy (Figure S6, Supporting Information). The endothermic breaking of hydrogen bonds makes ΔH_{mix} less negative and the entropy gain leads to the increase in the entropy term ($|T\Delta S_{\text{mix}}|$). **Once $|T\Delta S_{\text{mix}}|$ exceeds $|\Delta H_{\text{mix}}|$,**

ΔG_{mix} becomes positive and the entropy-driven phase separation occurs, wherein ILs aggregate into isolated domains (Figure 1d(2)). As a result, the IL domains cause severe light scattering at temperatures above the T_{cp} of the ionogel, causing the ionogel to become opaque. As the SPU_x polymers are crosslinked by multiple hydrogen bonds between the ASCZ moieties, the IL domains are well confined inside the ionogel, preventing the leakage of ILs (Figure 1d(3)). Below T_{cp} , the hydrogen bonds between ILs and the SPU_x polymers reform, leading to the miscibility of ILs and the SPU_x polymers. The IL domains contained within the ionogel diffuse evenly into the SPU_x network (Figure 1d(4)), restoring the original transparency of the ionogel.”

iii) Mingjie Liu Group at Beihang University

Finely Tuning the Lower Critical Solution Temperature of Ionogels by Regulating the Polarity of Polymer Networks and Ionic Liquids. CCS Chem. 2022, 4, 1386-1396.

“The LCST-type phase separation is a kind of phenomenon where the mixing Gibbs free energy (ΔG_{mix}) changes from negative to positive with increasing temperature. According to the pioneering work by Lee and Lodge,²⁸ it is believed that the coexistence of structure-forming solvatophobic and solvophilic groups in the polymer is necessary for LCST phase behavior. Similarly, in PBA/[C₂MIM][NTf₂] ionogel system, the solvophilic acrylate groups and the solvatophobic n-butyl side chains play important roles in the LCST phase behavior of ionogel (Figure 2a). **We speculate that the oriented solvation between the polymers and ILs caused by hydrogen-bonding effects and van der Waals interactions may serve as the driving force for the LCST phase behavior in our system.** Since hydrogen-bond interaction is stronger than van der Waals interaction, we supposed that the hydrogen bonds serve as the key factor for the LCST phase behavior in this system, which will be discussed later in this work. The van der Waals interaction here refers to the interactions between the polar domain of ILs and polar acrylate groups of the polymer backbone, as well as the interaction between nonpolar alkyl substituents on cation of ILs and the nonpolar alkyl side chain on polyacrylates network. The formation of polar and nonpolar domains in ILs is ascribed to Coulombic interactions³⁶ because the charged groups selectively solvate the charged groups and the uncharged groups (alkyl side chain of imidazolium cations) are expelled from this region as shown in Figure 2b. Herein, [C₂MIM][NTf₂] is a polar domain-dominated IL due to the short side chain on the imidazolium cation. Consequently, for the PBA/[C₂MIM][NTf₂] ionogel at a temperature below its LCST, it was supposed that ILs selectively solvate the solvophilic acrylate groups close to the polymer backbone, which results in the formation of a homogeneous and transparent

ionogel, as shown in Figure 1g (left). **The kinetics of thermoresponsive phase separation of ionogels with increasing temperature is described as follows. The hydrogen-bonds and van der Waals interactions between polymer networks and ILs are gradually weakened by heating. When the electrostatic interactions between ILs overwhelm the hydrogen-bonds and van der Waals interactions between polymer networks and ILs, the macroscopic phase separation occurs in the ionogel as shown in Figure 1g (right)."**

To facilitate your understanding of our proposed mechanism and the efforts we have made to address your concerns, the manuscript and relevant supplementary information are accompanied by the following:

"The microscopic interactions in thermosensitive dynamic phase separation were analyzed by in situ characterization and computational chemistry techniques. The temperature-dependent FTIR results (Fig. 3e) indicate a redshift of the characteristic peaks of ATI-B (from 1052 to 1050 cm^{-1} and 1348 to 1346 cm^{-1}), suggesting a weakening of the ion–dipole interaction between the IL and polymer⁵¹. This observation is strongly corroborated by density functional theory (DFT) calculation, which shows a significant decrease in the PBA-ion interaction energy per unit area (Supplementary Fig. 31). Likewise, the temperature-dependent Raman spectroscopy (TDRS) reveals a shift of the $\nu(-N-S)$ peak (Fig. 3f, Supplementary Fig. 32) to lower wavenumbers, implying enhanced Coulomb interaction between $[\text{Emim}]^+$ and $[\text{TFSI}]^-$ ⁵². This phenomenon is perfectly reflected in the molecular dynamics (MD) simulation, where the $[\text{Emim}]^+ - [\text{TFSI}]^-$ interaction energy increases with rising temperature and becomes dominant (Fig. 3g). Furthermore, two-dimensional correlation infrared spectroscopy (2DCOS FTIR) analysis suggests that the phase separation is triggered by the preferential dissociation of anions from the polymer, followed by cation re-association (Supplementary Fig. 33, Table 4 and Note 4). Taken together, these tightly coupled experimental and theoretical results provide compelling evidence for an entropy-driven dynamic phase separation mechanism above the LCST, wherein enhanced Coulomb forces promote the aggregation of solvent molecules detached from the polymer (Supplementary Note 5)⁵³.

Supplementary Note 2. LCST behavior of ATI-B.

The ATI proposed in this paper, particularly ATI-B, is a polymeric material with smart properties. The defining characteristic of thermoresponsive polymers lies in their response to temperature changes. Crucially, these polymers exhibit significant changes

in solubility at their phase transition temperatures, primarily due to the precise compatibility window within their binary polymer/solvent phase diagrams. Based on their thermoresponsive behavior, these polymers are primarily categorized into two types: low critical solution temperature (LCST) behavior and high critical solution temperature (UCST) behavior. LCST behavior implies that the polymer remains soluble in the solvent before reaching the phase transition temperature. Beyond this temperature, two distinct phases emerge—a diluted polymer phase and a concentrated polymer phase. Essentially, LCST characterizes the lowest point of phase diagram for a miscible system, beyond which only a single phase can exist, irrespective of polymer concentration (Supplementary Fig. 20a)¹³. In ionogel systems, the LCST phenomenon arises from the reversible interactions between the polymer and the solvent, governing the solvation state of the polymer. Differential scanning calorimetry (DSC) is a commonly used method for determining the LCST of thermosensitive polymers, as their phase transitions are often accompanied by endothermic events¹⁴. DSC measurements were conducted on ATI-B and ATI-BN. The results indicated that, besides a step-like shift in the baseline, ATI-B displayed an additional endothermic peak near 34 °C, which was absent in ATI-BN. The step-like shift corresponds to the glass transition temperature (T_g) of the polymer. Within this temperature region, the specific heat capacity of the polymer increases, leading to greater heat absorption and a shift of the baseline toward the endothermic direction. The weak endothermic peak is ascribed to the LCST behavior of the polymer (Supplementary Fig. 20b and 20c). At this temperature, the interactions between the polymer and the IL solvent are disrupted, leading to precipitation of the polymer from the solvent. This process is endothermic, producing a small enthalpy change. This is because in ATI-B, the dynamic variation in ion-dipole interactions between the polymer and ionic liquid is insufficiently large to be classified as a large-scale phase separation phenomenon accompanied by significant solvent expulsion (with large enthalpy change). Consequently, the endothermic peak on the DSC curve is not prominent, which is consistent with previous reports¹⁵. Moreover, a key characteristic of LCST behavior is a significant alteration in the material's light-scattering capability, aligning with our observations in visible-wavelength transmittance tests (Supplementary Fig. 16).

Supplementary Note 5. Entropy-driven dynamic phase separation in ATI-B.

Numerous studies report that hydrogen bonding and cation- π interactions can induce ordered structures between ILs and polymers, thereby generating a negative mixing

entropy (ΔS_{mix}). Similarly, we propose that oriented solvation driven by ion-dipole interactions in our system could serve as the driving force for the observed LCST behavior. At room temperature, the Gibbs free energy of mixing ($\Delta G_{mix} = \Delta H_{mix} - T\Delta S_{mix}$) is negative. However, heating endothermically disrupts these ion-dipole interactions, making ΔH_{mix} less negative. Concurrently, the system gains entropy, increasing the magnitude of the entropic contribution ($T|\Delta S_{mix}|$). When $T|\Delta S_{mix}|$ exceeds $|\Delta H_{mix}|$, ΔG_{mix} becomes positive. This results in entropy-driven phase separation, where the ILS segregate into discrete domains. ”

3. The correlation between formulation and performance is not well investigated. The authors only provide on the LCST values of different formulations. Some other important properties, including mechanical properties, low temperature transparency, high temperature transmittance, etc, should also be investigated, compared, and discussed. This will help readers better understand the materials design and proposed mechanisms.

Response: We sincerely appreciate the reviewer’s insightful and constructive comments. It is indeed crucial to investigate the correlation between formulation and performance. We fully agree that relying solely on LCST values is inadequate to comprehensively elucidate the material design and underlying mechanisms. In response to your suggestions, we have now systematically supplemented the experimental data on mechanical properties, low-temperature transparency, and high-temperature transmittance under different formulations, and have integrated comparative analyses and discussions into the revised manuscript.

“By varying the ratio of the polymer to the IL, the LCST of ATI-B can be tailored between 28°C and 41°C to meet specific requirements (Supplementary Fig. 21 and Table 2). Except for LCST, the formulation adjustments to the ATI-B will not significantly affect the mechanical properties or low-temperature transparency of ATIs (Supplementary Fig. 22 and Table 3).

Supplementary Figure 21. DSC curves of ATI-B in different polymer/IL ratios.

Supplementary Figure 22. Temperature dependence of transmittance at 550 nm for ATIs with different ATI-B formulations (same as Supplementary Table 2), insets showing transparency of ATIs at -70°C. Scale bar: 1 cm.

Supplementary Table 3. Mechanical properties of ATI under different ATI-B formulations (see Supplementary Table 2), including tensile strength, toughness, and Young's modulus.

Group	I	II	III	IV
Tensile strength (MPa)	5.21	5.46	6.18	5.35
Toughness (MJ/m ³)	15.6	17.6	20.2	22.3
Young's modulus (MPa)	6.75	6.80	8.25	6.12

The comprehensive mechanical properties of ATI-BN significantly outperform those of ATI-B, making it the primary contributor to the mechanical performance of ATI (see Section 4). Consequently, fine-tuning the ATI-B formulation has not been observed to affect the mechanical behavior of ATI, including tensile strength, fracture toughness, and Young's modulus. ”

We believe these additional experiments and extended discussions have reinforced the logical chain of “formulation–structure–performance” in the manuscript and made the conclusions more convincing. We are deeply grateful for your suggestions, which have prompted us to undertake these meaningful additions and greatly improved the quality of this work. We hope that these revisions and supplementary data meet your expectations.

The authors are requested to resolve these fundamental problems and provide more solid experimental data before it can be recommended for publication in Nature Communications.

Response: Thank you for your thorough evaluation of our manuscript and for raising these essential points. We believe that these revisions and additions have substantially enhanced the depth, reliability, and clarity of our study. We hope these efforts can clarify readers' questions about the proposed mechanisms to the greatest extent possible and to ensure that the manuscript meets the level of scholarly excellence expected by *Nature Communications*.

Once again, we are grateful for your constructive and discerning feedback, which has been valuable in improving this work. We hope that the revised manuscript now addresses all your concerns satisfactorily and merits your positive recommendation.

Reviewer #3 (Remarks to the Author)

The authors have fully addressed all my concerns and have improved the quality and robustness of the manuscript through these revisions. I believe the work is highly innovative and meets the standards for publication. I recommend the acceptance of this manuscript in its current form.

Response: Thank you for your positive and encouraging feedback. We are delighted that you find the revisions satisfactory and that the innovation of the work is now clearly demonstrated. We sincerely appreciate the time and expertise you have contributed throughout the review process; your constructive suggestions have been instrumental in enhancing the robustness of our manuscript.

Point-by-Point Response to Reviewers' Comments

Manuscript ID: NCOMMS-25-68161C

(Comments in black, responses in light blue, revised texts in *blue italics*,
important revision in *highlighted blue italics*)

We sincerely thank the editor and reviewers for dedicating additional time and effort to re-evaluate our work. Your comments have been invaluable in helping us improve the clarity and rigor of the manuscript. We have carefully considered all points and have revised the manuscript and supplementary information accordingly. Below, we provide a detailed, point-by-point response.

Reviewer #1 (Remarks to the Author)

The X-ray scattering/diffraction results show some ordered/crystalline structures, and the dimension of about 7 nm cannot cause opaque or translucent appearance of the hydrogels. The authors hypothesize that the 7 nm objects may aggregate to cause opaqueness. But no scattering data support this key hypothesis. I would like to recommend it to Nature Communications only when this important fundamental issue is supported by solid experimental data that verify the formation of phase separated structures.

Response: Thank you very much for your thorough evaluation and insightful comments on our manuscript. We sincerely apologize for any misunderstandings caused by unclear expressions in our original submission. Your feedback has been invaluable in helping us refine our work. We agree that solid experimental evidence for phase separation is paramount. We believe the concern stems from a conflation of the structures and functions of two independent layers in our asymmetric thermochromic ionogel (ATI). Below, we provide a point-by-point response to address your concerns. We first clarify this fundamental design to resolve the misunderstanding, and then present the direct experimental evidence for phase separation in the relevant layer.

1. Clarification of the distinct roles of two layers: the origin of the misunderstanding.

Our ATI is intentionally designed as a bilayer asymmetric system, where each layer has a distinct and independent function:

1) **The ATI-BN Layer (Mechanical Support Layer):** This is a highly transparent, tough layer. The characteristic ~ 7.3 nm feature obtained from SAXS (Supplementary Fig. 14) originates exclusively from this layer, not the thermoresponsive layer (ATI-B). Crucially, its size is far below the wavelength of visible light (380–780 nm), and thus, it does not contribute to optical scattering or opacity; ATI-BN remains highly transparent under all conditions. The in-situ phase separation in ATI-BN is a well-established toughening mechanism for soft materials, as documented in literature (e.g., Nat. Mater. 2022, 21, 359–365) and explained in Supplementary Fig. 12 and Fig. 14, where it enhances mechanical properties through a bicontinuous network.

2) **The ATI-B Layer (Optical Switching Layer):** This is the thermoresponsive layer that exhibits the transparent-to-opaque transition. The optical change is driven by a separate, micron-scale phase separation process that occurs solely within this layer.

In summary, the 7.3 nm structure (in ATI-BN) and the optical transition (in ATI-B) are completely unrelated. The request for “scattering data” to link the 7.3 nm feature to opacity is based on a premise that does not exist in our system. We have revised the manuscript text (Results, Figure Captions, and supplementary information) and added a schematic (Supplementary Fig. 1) to make this distinction unambiguous.

Supplementary Figure 1. The bilayer asymmetric structure consisting of ATI-B and ATI-BN within ATI, where each layer has a distinct and independent function.

Supplementary Figure 12. Schematic of phase separation in ATI-BN. *a)* Homopolymer networks of PBA and PNIPAM; and copolymer networks of P(BA-co-NIPAM) in phase-separated structures. *b)* Bicontinuous networks in in situ phase-separated structures.

Supplementary Figure 14. Analysis of bicontinuous structure in ATI-BN. *a)* 1D SAXS spectra. *b)* SEM image at the interface of ATI, with ATI-BN (rough) on the left and ATI-B (smooth) on the right. *c)* The schematic illustrates the bicontinuous structure and phase-separated correlation length in ATI-BN. Its characteristic size is about 7.3 nm, which is far smaller than the wavelengths of visible light, thereby ensuring the material remains transparent.

2. Experimental support for phase separation in the ATI-B layer and its role in transparency to opacity transition:

The reviewer rightly emphasizes the need for solid experimental verification of phase separation. We provide direct visual evidence for this process in the ATI-B layer.

In the original manuscript, we presented optical microscopy images (Fig. 3b) showing the emergence of micron-scale domains upon heating. To make this evidence more compelling, we have now performed additional in-situ temperature-controlled microscopy and captured a real-time video of the entire process (provided as Supplementary Movie 3).

- 1) Key frames extracted from this video are presented in the new Supplementary Fig. 17 (see figure below). They unambiguously show the formation, growth, and coalescence of ionic liquid-rich domains within the ATI-B layer as the temperature crosses the transition point.
- 2) The size of these domains evolves into the micrometer range (1-10 μm), which is commensurate with the wavelength of visible light. This is the direct cause of strong Mie scattering and the observed macroscopic opacity.
- 3) This optical microscopy evidence is a direct and established method for observing phase separation in soft materials, providing unambiguous proof of the formation of phase-separated structures.

Fig. 3: Dynamic phase separation principle and microscopic interaction analysis of ATIs. *a* Proposed schematic of the thermosensitive dynamic phase separation in ATI-B. *b* Optical microscopy images of ATI during the heating process. *c* Proposed schematic of the propagation path of light inside ATI-B below (left) and above (right) the LCST. *d* Optical photos of ATI under different temperatures. *e* Temperature-dependent FTIR spectra. *f* Temperature-dependent Raman spectra (colour scale: red, high intensity; blue, low intensity). *g* MD snapshots of PBA and [Emim][TFSI] at 298K and 313K. The calculated interaction energies of ion-dipole interactions (PBA-[Emim]⁺, PBA-[TFSI]⁻) and ion-ion interactions ([Emim]⁺-[TFSI]⁻) in ATI at 298K and 313K; insets show the ESP of the optimized structure of [Emim]⁺-[TFSI]⁻ at the two temperatures (colour scale: red, negative; blue, positive).

Supplementary Figure 17. Real-time observation of the thermally induced phase separation in the ATI-B layer. Selected frames from Supplementary Movie 3 showing the thermal evolution of micron-scale ionic liquid domains at the critical transition temperature. Scale bar: 100 μm .

3. Supporting Molecular-Scale Evidence for the Phase Separation Mechanism.

At the molecular level, WAXS characterization of ATI-B (Fig. 2d-ii, Supplementary Fig. 15 and Table 1) reveals changes in ion cluster dimensions, reflecting the initial stages of phase separation at the ionic scale (e.g., entropy-driven contraction of ion clusters). While these WAXS features are too small to directly cause opacity, they provide critical insights into the early dynamics of ion aggregation, which escalate to macroscopic phase separation.

In summary, we have taken the following actions to address the reviewer's concern:

- 1) Clarified the Text: Revised the manuscript throughout to explicitly and repeatedly distinguish the functions of the ATI-BN (mechanical) and ATI-B (optical) layers.
- 2) Enhanced Visualization: Added Supplementary Movie 3 and a corresponding figure with key frames (Supplementary Fig. 17) to dynamically demonstrate the phase separation process in the ATI-B layer.
- 3) Added a Schematic: Included Supplementary Fig. 1 to schematically illustrate the bilayer asymmetric structure and their independent functions.
- 4) Updated Captions: Revised the caption of Supplementary Fig. 14 (SAXS) to explicitly state: "The schematic illustrates the bicontinuous structure and in-suit phase-separated correlation length in ATI-BN. The characteristic size of separated phase is about 7.3 nm, which is far below the wavelengths of visible light, thereby strengthening the ionogel without changing its transparency."

We believe that the direct visual evidence from microscopy (Fig. 3b and new Supplementary Movie 3 and Fig. 17), supported by molecular-scale insights from WAXS, provides the solid experimental foundation for the phase separation that drives the optical switch. This evidence, combined with the clarified manuscript text and supplementary information, fully addresses the core of the reviewer's concern.

"The in situ and dynamic phase separation in ATIs were analyzed at the microscopic level using small-angle X-ray scattering (SAXS) and wide-angle X-ray scattering (WAXS) techniques. The SAXS results show that ATI-B is highly homogeneous on the 1-50 nm scale, and the absence of scattering rings in the 2D SAXS spectrum proves that there is almost no internal phase separation at room temperature (Fig. 2d-i). In contrast, ATI-BN exhibits more pronounced scattering rings, and its SAXS spectrum displays distinct step-like peaks. Through computational fitting, its phase separation size was determined to be approximately 7.3 nm (Supplementary Fig. 13)⁴⁰. This size is far below

the visible light wavelength range, so ATI-BN remains highly transparent. The thermally induced transparent-to-opaque transition of ionogels is solely related to ATI-B. The variable-temperature WAXS results of ATI-B reveal that the characteristic dimensions of its scattering peaks shift from 4.5 Å and 7.7 Å to 4.6 Å and 7.2 Å (Fig. 2d-ii, Supplementary Fig. 14 and Table 1). This change can be attributed to the increase in polymer chain spacing due to thermal expansion effects, coupled with the entropy-driven contraction of [Emim]⁺–[TFSI]⁻ ion clusters⁴¹, which is a precursor to microscale phase separation.

Fig. 2: Preparation, composition, and characterization of asymmetric structures of ATIs. **a** ESP simulations of key ATI components: polymer monomers BA and NIPAM, and IL solvent [Emim][TFSI]. **b** FTIR spectra of solvent-free PBA, pure IL, and ATI-B. **c** Microscopic image (left) and SEM image (right) of the cross-section of the ATI, distinctly showing its asymmetric structure composed of ATI-BN and ATI-B. **d** 2D spectra of SAXS (i) and WAXS (ii). **e** Raman spectra of ATI-B and ATI-BN; insets i, ii, iii are Raman mapping images of ATI-B, ATI-BN, and ATI cross sections, respectively. **f** Optical images of physical bilayer ionogel and ATI (upper PBA ionogel layer, stained with metanil yellow; lower P(BA-co-NIPAM) ionogel layer, stained with brilliant green) and schematic of the proposed interface. **g** Characterization of the rheological properties of ATIs, including amplitude sweeping (left), frequency sweeping (middle), and temperature ramping (right).

Supplementary Figure 15. 1D WAXS spectra of ATI-B.

Supplementary Table 1. *d*-value in the variable-temperature WAXS of ATI-B.

	20°C	30°C	40°C
d-1	7.7 Å	7.6 Å	7.2 Å
d-2	4.5 Å	4.5 Å	4.6 Å